



# Tracing devastating fires in Portugal to a snow archive in the Swiss Alps: a case study

Dimitri Osmont[1,2,3,a], Sandra Brugger[3,4,*], Anina Gilgen[5,*], Helga Weber[3,6,*], Michael Sigl[1,3], Robin L. Modini[7], Christoph Schwörer[3,4], Willy Tinner[3,4], Stefan Wunderle[3,6], Margit Schwikowski[1,2,3]

[1]Laboratory of Environmental Chemistry, Paul Scherrer Institut, 5232 Villigen, Switzerland
[2]Department of Chemistry and Biochemistry, University of Bern, 3012 Bern, Switzerland
[3]Oeschger Centre for Climate Change Research, University of Bern, 3012 Bern, Switzerland
[4]Institute of Plant Sciences, University of Bern, 3012 Bern, Switzerland
[5]Institute for Atmospheric and Climate Science, ETH Zürich, 8092 Zürich, Switzerland
[6]Institute of Geography, University of Bern, 3012 Bern, Switzerland
[7]Laboratory of Atmospheric Chemistry, Paul Scherrer Institut, 5232 Villigen, Switzerland
*These authors equally contributed to this work.
[a]Now at Institut des Géosciences de l'Environnement, Université Grenoble-Alpes, 38400 Saint Martin d'Hères, France

*Correspondence to*: Margit Schwikowski (margit.schwikowski@psi.ch)

**Abstract.** Recent large wildfires, such as those in Portugal in 2017, have devastating impacts on societies, economy, ecosystems and environments. However, wildfires are a natural phenomenon, which has been exacerbated by land use during the past millennia. Ice cores are one of the archives preserving information on fire occurrences over these timescales. A difficulty is that emission sensitivity of ice cores is often unknown, which constitutes a source of uncertainty in the interpretation of such archives. Information from specific and well-documented case studies is therefore useful to better understand the spatial representation of ice-core burning records. The wildfires near Pedrógão Grande in Central Portugal in 2017 provided a test bed to link a fire event to its footprint left in a high-alpine snowpack considered a surrogate for high-alpine ice-core sites. Here, we (1) analyzed black carbon (BC) and microscopic charcoal particles deposited in the snowpack close to the high-alpine research station Jungfraujoch in the Swiss Alps, (2) calculated backward trajectories based on ERA-Interim reanalysis data and simulated the transport of these carbonaceous particles using a global aerosol-climate model, and (3) analyzed the fire spread, its spatial and temporal extent, as well as its intensity, with remote sensing (e.g. MODIS) active fire and burned area products. A peak of atmospheric equivalent BC (eBC) observed at the Jungfraujoch research station on 22nd June, with elevated eBC levels until the 25th June, is in correspondence with a peak in refractory BC (rBC) and microscopic charcoal observed in the snow layer. rBC was mainly scavenged by wet deposition and we obtained scavenging ratios ranging from 81 to 91. Unlike for microscopic charcoal, the model did not well reproduce the observed rBC signal. Our study reveals that microscopic charcoal can be transported over long distances (1500 km), and that snow and ice archives are much more sensitive to distant events than sedimentary archives, for which the signal is dominated by local fires. Microscopic charcoal concentrations were exceptionally high since this single outstanding event deposited as many charcoal particles per day as during an average year in ice cores. This study unambiguously links the fire tracers in the snow



with the highly intensive fires in Portugal, where a total burned area of 501 km$^2$ was observed on the basis of satellite fire
products. According to our simulations, this fire event emitted at least 203.5 tons of BC.

## 1 Introduction

Fires are an important component of terrestrial ecosystems as they substantially control vegetation cover and contribute to
the global carbon budget (e.g. Bond et al., 2005; Hantson et al., 2015). Current global $CO_2$ emissions from fires, including
landscape and biomass, represent around 50 % of the global $CO_2$ emissions produced by fossil fuel burning (Bowman et al.,
2009). Fires exert an influence on the climate system as they emit greenhouse gases and aerosols (Andreae and Merlet, 2001)
and change surface albedo (Randerson et al., 2006), as well as on vegetation and soil carbon (Page et al., 2002). In the last
two decades, the occurrence of devastating wildfires has increased in many regions of the world, leading to substantial
socioeconomic and environmental consequences (Moritz et al., 2014). In the context of global warming, fire risk (Pechony et
al., 2010), frequency (Keywood et al., 2013), and season severity (Flannigan et al., 2013) are potentially increasing with
significant feedbacks on terrestrial and atmospheric systems (Bowman et al., 2011). To understand these future impacts,
paleofire reconstructions provide an important tool for assessing long-term changes in past fire activity and can help to
disentangle the influence of climate and humans on biomass burning. Most of the currently available sedimentary records
have large chronological uncertainties in the youngest part of the records (Marlon et al., 2016) and reflect small local to
regional catchments (Adolf et al., 2018). To address global fire activity trends, ice cores from polar (see e.g. Arienzo et al.,
2017; Fischer et al., 2015; Keegan et al., 2014; Legrand et al., 2016; Zennaro et al., 2014) and high-altitude glaciers (see e.g.
Brugger et al., 2018a, 2019a; Eichler et al., 2011; Osmont et al., 2018, 2019; Yalcin et al., 2006) have a high potential since
they record fire activity from regional to continental scales and usually provide well-constrained chronologies (see e.g.
Herren et al., 2013; Konrad et al., 2013; Uglietti et al., 2016).
Fires emit a wide range of chemical compounds and particles to the atmosphere, such as black carbon (BC) and charcoal. If
transported over long distances, these particles can be deposited with precipitation or by gravitational settling on the
snowpack, where they will be archived, and can be subsequently retrieved by ice-core drilling. However, ice-core catchment
areas are not often precisely known due to a lack of information about emission, transport and deposition processes. Case
studies are essential to understand these processes including fire extent, fuel load, plume transport, biomass burning tracer
deposition and preservation in the snowpack (e.g. Kaspari et al., 2015). Modelling of fire tracers (e.g. BC or charcoal) from
the fire source to the deposition site provides a direct link and quantification of the archived biomass burning tracers.
Detailed information to quantify the amount, frequency and intensity of biomass burning emissions relies mainly on satellite
observations (e.g. occurrence of active fires, fire radiative power (i.e. intensity; FRP), burned area and vegetation cover). A
limiting factor is the availability of long-term satellite products with sufficient temporal and spatial resolution. Recently, an
active fire product was developed for Europe reaching back until 1985 (see Weber and Wunderle, 2019). Therefore, a direct
linkage between specific fire events and ice core observations is very challenging, especially for biomass burning events that





occurred some decades or centuries ago. Thus, the fire footprint of ice-core sites and the preservation of single biomass burning events in these archives remain largely unknown. Pioneer case studies have shown that elevated concentrations of ammonium, potassium and formate in Greenland's atmosphere and snow could be directly linked to forest fires in Canada (Dibb et al., 1996). More recent case studies confirmed that BC emissions from fires can be preserved in snow and ice

archives at sub-continental scales. For instance, BC peaks in Greenland snowpits were associated with specific biomass burning events in Canada with the help of remote sensing and modelling tools (Thomas et al., 2017). BC emissions from the oil well fires in Kuwait in 1991 during the Gulf War were detected unambiguously in an ice core from Muztagh Ata, Northern Tibet (Zhou et al., 2018). However, for Europe, characterized by highly fragmented landscapes and smaller mean burned area compared to other continents (Mouillot and Field, 2005), specific case studies are missing.

Here, we focus on an outstanding wildfire event starting on 17[th] June 2017 in Portugal. Portugal was affected by a severe heat wave with temperatures above 40 °C in June 2017 and during a dry thunderstorm, lightning ignited the forests dominated by non-native *Eucalyptus* plantations near Pedrógão Grande in Central Portugal (Fig. 1). The highly flammable vegetation resulted in a rapid and uncontrolled spread of the fire with devastating impacts, leading to the worst death toll Portugal ever experienced for a fire. 64 people lost their life and 254 were injured (Gómez-González et al., 2018) until the

fires were finally extinguished on 24[th] June. A large plume of smoke was emitted and transported to the northeast from the burning site. On 21[st] June, the automatic lidar operated by MeteoSwiss (Federal Office of Meteorology and Climatology, Switzerland) in Payerne, Switzerland, detected a layer of smoke between 3000 and 5000 m a.s.l. corresponding to the arrival of the plume. The signal intensified on 22[nd] June and its Portuguese origin was confirmed by atmospheric backward trajectory analysis (MeteoSwiss, 2017). The smoke layer became visible in the morning hours of 22[nd] June on the

Jungfraujoch (JFJ) research station webcam, located at 3580 m a.s.l. in the Bernese Alps. At the same time, a peak in atmospheric equivalent black carbon (eBC, following the terminology recommendations by Petzold et al., 2013) was detected by the Multi-Angle Absorption Photometer (MAAP, optical method for BC quantification, Petzold and Schönlinner, 2004) installed at the research station (Fig. 2a). Elevated values lasted until 25[th] June when the first snowfall after the event occurred.

The exceptional conditions of this Portuguese fire event producing a plume that was recorded unambiguously at the high-alpine research station JFJ in Switzerland, located close to perennial snow archives, provided an ideal situation to analyze in detail the processes related to emission, transport and deposition of fire tracers in high-mountain snowpack. Several well-studied ice cores were already collected in the vicinity, namely from Fiescherhorn glacier and Colle Gnifetti, which are located 6 km east and 70 km south, respectively (Jenk et al., 2006; Sigl et al. 2018).



## 2 Methods

### 2.1 Study site and meteorological conditions

The high-altitude research station Jungfraujoch (46°32´ N 7°59´ E, Fig. 1) is located on the eponymous pass between the summits of Jungfrau and Mönch in the Bernese Alps. Built at an altitude of 3580 m a.s.l., the surrounding high-alpine environment is characterized by glaciers and cliffs. The north side, facing the Swiss Plateau, consists of a steep ice fall while the relatively flat south slopes host the large Jungfraufirn glacier feeding the Aletsch glacier. The JFJ site lies only partially within the free troposphere, being frequently influenced by uplifted air from the planetary boundary layer (e.g. Baltensperger et al., 1997; Bukowiecki et al., 2016). Aerosol monitoring at JFJ provides continuous long-term atmospheric records of many parameters including eBC concentration. In addition, a weather station provides meteorological data, but precipitation is not monitored at JFJ due to the difficulty of obtaining accurate data as most precipitation occurs in form of snow in frequent association with strong winds. The closest precipitation data available is from the automatic weather station Itramen (2162 m a.s.l.), operated by the Institute for Snow and Avalanche Research (WSL-SLF) and located 9 km to the northwest of JFJ. Following the Portugal fires, precipitation (rainfall) was recorded at Itramen on 25[th], 26[th], 27[th], 28[th] and 29[th] June, bringing 32, 42.8, 2.0, 49.6 and 35.2 mm of water, respectively (data from SLF © 2019, SLF). We used the JFJ webcam to determine whether snowfall occurred at the same time at JFJ, which was the case except for 27[th] June.

### 2.2 Snowpit and sampling

A snowpit was collected on 30[th] June 2017, around 20 m off the prepared trail between JFJ and Mönchsjochhütte and 400 m east from the Jungfraujoch tunnel exit onto the glacier, at an altitude of 3460 m a.s.l. A massive ice layer prevented us from digging deeper than 1.10 m. Density was measured on the spot for each layer, by weighing a stainless steel cylinder of known volume filled with snow. Following the stratigraphic study, 20 samples were collected at about 5 cm resolution by pushing pre-cleaned 50-mL polypropylene vials vertically in the snow wall. Ice layers were sampled specifically (if thick enough) to check the potential impact of melting on the chemical composition of the snowpack. 20 replicate samples were retrieved at the same resolution to test the reproducibility of the experiment. In addition, 6 pre-cleaned 1-L PETG jars were filled with 0.2 to 0.5 kg snow for microscopic charcoal analysis, at 10 cm resolution between 20 and 80 cm depth.

### 2.3 Analytical methods

Samples were stored in frozen state and were melted just before analysis of refractory BC (rBC) and major ions at the Paul Scherrer Institute. rBC analysis followed the method described by Wendl et al. (2014): after sample melting at room temperature and a 25-min sonication in a ultrasonic bath, rBC was quantified using a Single Particle Soot Photometer (SP2, Droplet Measurement Technologies, USA, Schwarz et al., 2006; Stephens et al., 2003) coupled to an APEX-Q jet nebulizer (Elemental Scientific Inc., USA). Further analytical details regarding calibration, reproducibility and autosampling method can be found in Osmont et al. (2018). The samples were subsequently analyzed for 13 ions (5 cations: ammonium $NH_4^+$,



calcium $Ca^{2+}$, magnesium $Mg^{2+}$, potassium $K^+$ and sodium $Na^+$; and 8 anions: acetate $CH_3COO^-$, chloride $Cl^-$, fluoride $F^-$, formate $HCOO^-$, methanesulfonate (MSA) $CH_3SO_3^-$, nitrate $NO_3^-$, oxalate $C_2O_4^{2-}$ and sulfate $SO_4^{2-}$) by ion chromatography (850 Professional I*C*, Metrohm, Switzerland).

To estimate microscopic charcoal concentrations, we added a known amount of *Lycopodium* marker spores to the six snow samples dedicated to palynological analyses, and prepared the samples following the evaporation-based protocol for ice samples developed by Brugger et al. (2018b). Microscopic charcoal particles were identified as completely opaque particles with a major axis >10 μm and with angular shape following Tinner and Hu (2003). We counted to a combined sum of 200 items (microscopic charcoal + *Lycopodium* spores; Finsinger and Tinner, 2005).

## 2.4 Air mass trajectories and transport simulations

To link the observations in the snowpack with the fire emissions in Portugal, we calculated three day backward trajectories at a 6-hourly resolution with the Lagrangian analysis tool LAGRANTO (Sprenger and Wernli, 2015) based on ERA-Interim reanalysis data (Dee et al., 2011). The trajectories started at the coordinates of JFJ, at 20 equidistant levels in pressure coordinates between 700 hPa and 200 hPa.

We used the global aerosol-climate model ECHAM6.3-HAM2.3 to simulate the transport of BC and microscopic charcoal
(Gilgen et al., 2018; Stier et al., 2005; Zhang et al., 2012). The large-scale wind velocities of the simulations were nudged towards ERA-Interim reanalysis data. Two simulations were conducted: i) a simulation without any fire emissions and ii) a simulation including the fire emissions that occurred in Central Portugal (−8.45° to −7.85° E; 39.75° to 40.25° N) in June. Daily fire emission fluxes were calculated from the Fire Inventory from NCAR (FINN version 1.6.; Wiedinmyer et al., 2011) and interpolated to the model grid. For anthropogenic aerosol emissions, we used ACCMIP (Atmospheric Chemistry and
Climate Model Intercomparison Project) interpolated emissions for the year 2008 (Lamarque et al., 2010), being constant over the year. A spatial resolution of T127L95 was applied (≈100 km at the equator). To simulate the microscopic charcoal particles, we chose a density of 0.6 g cm$^{-3}$, a geometric mean radius upon emission of 5 μm, a threshold radius of 4.9 μm and a scaling factor of BC mass emissions of 250. These parameters were extensively tested, compared and validated by Gilgen et al. (2018).

## 2.5 Satellite observations

To analyze the fire source itself, i.e. the spatial and temporal extent as well as the intensity of the burning near Pedrógão Grande, we used different fire products retrieved from satellite imagery over the study area. The fires were detected by several satellite sensors and during multiple overpasses (e.g. Visible Infrared Imaging Radiometer Suite (VIIRS), Moderate Resolution Imaging Spectroradiometer (MODIS) or Advanced Very High Resolution Radiometer (AVHRR)), which allows
to study the temporal evolution (i.e. progression) of the fires. Here, we chose the Thermal Anomalies & Fire Daily L3 Global Product, which is also utilized for the FINN emission model, together with the Burned Area Monthly L3 Global Product of MODIS.



The MODIS sensors on board NASA's Earth Observing System Terra and Aqua satellites have a return period of one to two days with a daytime equator crossing time at 10:30 am (1:30 pm) for Terra (Aqua). The areal extent of the study site was defined as −8.45° to −7.85° longitude and 39.75° to 40.25° latitude according to the location of the forest fires and also used for the modelling (see Section 2.4). For June 2017, we downloaded the respective tiles over the study area of both products: i) the 1 km daily thermal anomaly and fires MOD14A1 (MODIS/Terra) and MYD14A1 (MODIS/Aqua: both V006; Giglio et al., 2015a) products and ii) the monthly 500m Burned Area Product MCD64A1 (V006; Giglio et al., 2015b). The latter is a combined product of the burned area detected by the MODIS sensors onboard the Terra and Aqua satellites. It was resampled to a spatial resolution of 1 km. The tiles of the MOD14A1/MYD14A1 products were mosaicked into one image. We lastly reprojected the scenes of both products accordingly to the FINN emission data (version 1.6; Wiedinmyer et al., 2011). Both products were analyzed in terms of active fire progression, fire radiative power (FRP), burned area date and extent, and cloud coverage. Moreover, we compared them with the spatial distribution of emission species, like BC, $NO_x$ and $PM_{2.5}$, derived from the FINN database.

## 3 Results and discussions

### 3.1 Snowpit profile

The snowpit profile (Fig. 2b) showed five layers with different grain size and density (Fig. 2c). Layer A, from 0 to 10 cm depth (d = 0.2 g cm$^{-3}$), was composed of fresh and very light powder snow corresponding to the snowfall on 29$^{th}$ June. Layer B, from 10 to 22 cm (d = 0.37, g cm$^{-3}$) was made of light snow probably originating from the snowfall on 28$^{th}$ June. Snow in layer C, from 22 to 50 cm (d = 0.54 g cm$^{-3}$), already experienced some transformation as bigger (2–3 mm) round-shaped grains were observed that could relate to the snowfalls that occurred on 25$^{th}$ and 26$^{th}$ June. For those days, a good agreement was obtained between the precipitation amount from Itramen and the snow layer height corrected for density (Fig. 2a), confirming our time attribution. Below 50 cm depth, accurate dating becomes impossible, but the frequent presence of ice layers and more compact snow indicates melting due to warmer temperatures. This is in line with the June 2017 heat wave in Switzerland that lasted from 19$^{th}$ to 24$^{th}$ June (MétéoSuisse, 2017). Layer D, from 50 to 76 cm depth, was composed of denser and compact snow (d = 0.55 g cm$^{-3}$). Lastly, below 78 cm depth, layer E seemed even more compact, although the density did not significantly change (d = 0.52 g cm$^{-3}$).

### 3.2 Fire tracers: rBC, microscopic charcoal and major ions

A remarkable peak with concentrations up to 9.8 ng g$^{-1}$ is visible in the rBC profile (Fig. 2d) from 22 to 37 cm depth (samples 4 to 6; layer C), corresponding to the first snowfalls recorded after the event. This suggests that atmospheric BC was probably scavenged by snow on 25$^{th}$ and 26$^{th}$ June, which is in agreement with a drop in atmospheric eBC concentration observed simultaneously (Fig. 2a). Wet deposition seems to be the preferential pathway as the rBC peak is spread over the whole accumulated snow layer while dry deposition would rather create a thin and highly concentrated layer. Several studies



indicate that rBC is mainly scavenged from the atmosphere via wet deposition processes (Cape et al., 2012; Ruppel et al.,
2017; Sinha et al., 2018). The uppermost two layers (A–B, samples 1 to 3) show very low rBC concentrations (average: 0.21
ng g$^{-1}$), in agreement with the clean atmospheric conditions that prevailed on 28$^{th}$ and 29$^{th}$ June with eBC concentrations
below 10 ng m$^{-3}$. Below the rBC peak, from 37 to 110 cm depth, rBC concentrations with an average of 2.0 ng g$^{-1}$ show
some variability but no clear trend or peak. A very good agreement is obtained between the two series of replicate rBC
samples (r = 0.90).

The microscopic charcoal record (Fig. 2e) shows a smaller peak at 60–70 cm depth (4000 fragments L$^{-1}$), and the main
maximum at 30–40 cm depth, reaching a factor of 10 higher concentrations (20 000 fragments L$^{-1}$) compared to the average
of all six samples. We relate the main peak to the fire event in Portugal. Compared to the rBC peak, the narrower
microscopic charcoal peak could indicate either a more important contribution of dry deposition versus wet deposition or a
more efficient scavenging during snowfall, possibly due to its larger particle size (> 10 μm major axis for microscopic
charcoal vs. < 1 μm diameter for rBC). Microscopic charcoal is a specific proxy for biomass burning, whereas rBC is less
specific and can also originate from fossil fuel combustion. The simultaneous rBC and microscopic charcoal peaks (rBC
samples 4 to 6 and microscopic charcoal sample 2, respectively) provide evidence for a common biomass burning source and
transport within the plume. No ice layer was observed in the uppermost 50 cm of the snowpit, which indicates that both rBC
and microscopic charcoal profiles were not affected by melting processes.

The rather large diameter of the rBC particles, with a mean mode of the rBC mass size distribution for the broadband high
gain (BBHG) channel of the SP2 of 306 nm and 291 nm for the two series of replicates, respectively (Fig. 2d), also suggests
a predominant biomass burning origin. This is in agreement with Lim et al. (2017), who observed a mean mass mode
diameter of 290.8 nm in the summer layers of the Elbrus ice core and attributed the size increase compared to winter layers
to the predominance of forest fires and/or agricultural fires as rBC source in summer. The mass size distribution remains
fairly similar throughout the profile and does not display higher values in the samples containing rBC peak concentrations.
This indicates that rBC size distributions in the falling snow are rather stable, as already observed by Sinha et al. (2018).
Only sample 13 shows a bigger fraction of large rBC particles, in association with a small peak in rBC and charcoal
concentration, potentially suggesting more local burning sources. On the contrary, no shift in the rBC size distribution is
visible for samples 4 to 6 during the peak in rBC and charcoal concentrations.

Continental-scale calibrations comparing satellite-based fire incidence and charcoal influx into lake sediment or peat bogs
suggest that microscopic charcoal particles > 10 μm mainly originate from regional sources with an average radius of ca. 40
km (Adolf et al., 2018). However, previous studies of charcoal in ice cores indicate that charcoal can be transported over
distances larger than 500 km (e.g. Brugger et al., 2018a; Eichler et al., 2011; Reese et al., 2013, Hicks and Isaksson, 2006).
These differences can be explained by dissimilar environmental conditions, as lake sediment and peat bog studies are usually
performed at low-elevation sites surrounded by potential burning sources while snowpits and ice cores originate from high-
latitude or high-altitude sites remote from vegetation (Brugger et al., 2018a, 2019b). These remote sites frequently lie within
the free troposphere: for JFJ, free troposphere background conditions are observed about 39% of the time, with a maximum



of 60% in winter and a minimum of 20% in summer (Bukowiecki et al., 2016). However, a travel distance from Portugal to JFJ of around 1500 km seems exceptional. Given the strong intensity of this fire (see Section 3.5), we hypothesize that

convection lifted a large amount of microscopic charcoal particles high up in the atmosphere enabling the long-distance travel. The microscopic charcoal peak concentrations observed in this study are outstandingly high and only comparable to exceptional peak events in other ice cores (Brugger et al., 2018a; Eichler et al., 2011; Reese et al., 2013). This is in contrast to rBC, for which such peak concentrations have been observed in the preindustrial part of European high-altitude ice cores, when rBC mainly originated from biomass burning (Lim et al., 2017; Sigl et al., 2018).

In addition to rBC and charcoal, ions such as ammonium ($NH_4^+$), formate ($HCOO^-$), acetate ($CH_3COO^-$) and nitrate ($NO_3^-$) have been used as fire tracers (Arienzo et al., 2017; Fischer et al., 2015; Legrand et al., 2016; Savarino and Legrand, 1998), although other emission sources exist, such as direct biogenic emissions (ammonium, formate), oxidation from volatile organic compounds (formate, acetate) and anthropogenic sources (agriculture for ammonium, traffic and agriculture for nitrate). Except for a few values, a good agreement was obtained between the two sets of replicates of the major ion profiles,

particularly for ammonium and nitrate (Fig. 3). Surprisingly, these ions do not display any significant enhancement of their concentration from 22 to 37 cm depth, showing that, under present day conditions, even an exceptional fire event does not necessarily result in an exceptional peak of ammonium, formate, acetate or nitrate. This is mainly the result of the predominance of other emission sources (biogenic and anthropogenic emissions). To a lesser extent, varying atmospheric lifetimes compared to rBC and microscopic charcoal, and different sensitivities to wet deposition (different scavenging

ratios) might also explain these differences. Among the ionic species, only calcium shows a concomitant peak with rBC and microscopic charcoal. One possible explanation is that snowfalls on 25th and 26th June were the first ones following the June 2017 heat wave in Switzerland. Dust concentrations might have been elevated during this dry and warm period and those subsequent snowfalls could have cleaned the atmosphere and led to wet deposition of dust-related ions such as calcium.

### 3.3 rBC scavenging ratios

rBC scavenging ratios (W) were calculated to determine the total scavenging of rBC from air to snow by snowfall, based on the following formula: $W = \rho C_s/C_a$, with $\rho$ the air density in g m$^{-3}$, $C_s$ the concentration of rBC in snow in ng g$^{-1}$ and $C_a$ the concentration of rBC in air in ng m$^{-3}$ (Schwikowski et al., 1995). For $C_a$, we considered the daily average eBC concentration on 24th June, just before the precipitation starts, while for $C_s$, we used the average of samples 4–6 of the snowpit, corresponding to the days 25th and 26th June when peak values are found. The air density at JFJ of 842 g m$^{-3}$ was calculated

with the ideal gas law using a temperature of 273.15 K and a pressure of 66000 Pa obtained from MeteoSwiss for the JFJ weather station.

In our case, the calculation of rBC scavenging ratios (Table 1) is highly dependent on the choice of a mass absorption coefficient (MAC) for converting the light absorption intensity given by the MAAP into an eBC concentration. The objective is that the eBC mass concentrations from the MAAP precisely match the rBC mass concentrations from the SP2, in order not

to introduce a methodological bias as two different quantification methods are used. Previous studies suggested a median



MAC of $10.2 \pm 3.2$ m$^2$ g$^{-1}$ for JFJ (Liu et al., 2010) or $11.1 \pm 0.2$ m$^2$ g$^{-1}$ in summer (Cozic et al., 2008). Higher values ($13.3 \pm 3.0$ m$^2$ g$^{-1}$) have also been reported in the case of eBC from a purely biomass burning origin (Schwarz et al., 2008). An upper limit estimate of 20 m$^2$ g$^{-1}$ for the MAC was obtained during the CLACE 2016 summer campaign at JFJ (Motos et al., 2019). This value is of particular interest as both a MAAP and a SP2 were simultaneously sampling the air at JFJ during this

campaign. By applying a MAC of 20 m$^2$ g$^{-1}$, a perfect agreement could be obtained between the BC mass concentrations given by the MAAP and those given by SP2 (measurements not shown). Therefore, MACs of 10 and 20 m$^2$ g$^{-1}$ were chosen here ($\lambda = 637$ nm) to test the lower and upper limits, leading to a range of scavenging ratios from 41 to 91 (Table 1).

Few BC air-to-snow scavenging ratio values are available in the literature, usually ranging from 100 to 150 (Table 2). By comparing our scavenging ratios obtained at JFJ with values from previous studies, it appears that the choice of a MAC

value of 20 m$^2$ g$^{-1}$ seems preferable. Discrepancies can arise from the different locations implying various climatic conditions, from the use of different methods to quantify BC (light absorption, thermal-optical or incandescence), and from the different relative contribution of wet and dry deposition processes, depending on the location and elevation, with higher scavenging ratios at higher altitudes (Gogoi et al., 2018).

### 3.4 Atmospheric transport

The 3-day air mass backward trajectory analyses suggest Portugal as a very likely source for the atmospheric eBC (Fig. 4). For all the days between the 22$^{nd}$ and the 25$^{th}$ June, when atmospheric as well as deposited eBC concentrations were large, part of the air masses originated from Portugal, which is also supported by simulations from MeteoSwiss (MeteoSwiss, 2017). Simulations with ECHAM-HAM show no clear difference for BC when we compare the simulation with fires to the simulation without fires (Fig. 5). In our simulations, BC is preferentially removed by wet scavenging. This suggests that the

simulated BC deposition peak observed from 26$^{th}$ to 28$^{th}$ June is mainly reflecting the precipitation pattern, as precipitation frequently occurred at JFJ from June 25$^{th}$ on (Fig. 2a), and not a change in emission sources. Furthermore, other BC sources than fires seem to dominate in our simulations. This could be due to the relatively low spatial resolution, which smooths the topography, as well as due to the efficient vertical mixing in the lower troposphere in the model. As a consequence, overestimated levels of BC from regional anthropogenic sources can probably reach the location of JFJ in the model

compared to reality, and therefore mask the BC peak related to the Portuguese fires.

Microscopic charcoal has no other sources than biomass burning in our model. We observe a clear peak in the deposited microscopic charcoal fluxes on the 23$^{rd}$ of June (Fig. 5), which is predominantly caused by dry removal processes (i.e. dry deposition and gravitational settling) and near the observed peak on the 22$^{nd}$ of June for atmospheric eBC (Fig. 2a). The simulated deposition fluxes are still rather high on the 25$^{th}$ of June, when the observed peak in microscopic charcoal starts

(Fig. 2e). An exact temporal match cannot be expected due to the daily resolution of the fire emissions and the rather low spatial resolution of the model together with the dating uncertainties of the snow pit. Nevertheless, this result qualitatively confirms the hypothesis that the microscopic charcoal particles observed at JFJ originated from Portugal.



### 3.5 Observations by satellites

Quantification of the occurrence (active fires), intensity (maximum FRP) and extent (burned area) provided by satellite
products allows the modelling of biomass burning emissions. Here, we analyzed the temporal and spatial extent as well as
the FRP of the wildfires with the MODIS thermal anomaly and fire (MOD14A1/MYD14A1) and the MCD64A1 burned area
products together with emission species of the FINN v1.6. database. The fires evolved on the 17[th] June 2017 in the afternoon
(Fig. 6a) in a central location between the fire clusters. Later during the day, several fires were already burning according to
the burned area product, which indicates a fast spread in south- and northward direction. Clouds obscured observation by the
MODIS sensor on the 18[th] June. Therefore, fewer active fires and lower maximum FRP were detected on this day, resulting
in gaps in the MOD14A1/MYD14A1 product (Fig. 6a). However, the MCD64A1 burned area product shows the day by day
evolution of the fires, forming two big fires clusters. The maximum fire activity and vegetation consumption was observed
on the 19[th] and 20[th], as indicated by the burned area and the high maximum FRP values. Less powerful fires on the south and
mainly north edges of the two fire clusters burned from the 20[th] to 22[nd] until the fires were completely extinguished on the
24[th] of June.

The total area burned over these days accumulates to 501 km$^2$ according to the burned area product. The total area might be
underestimated by this product compared to the maximum FRP, which indicates a larger spatial extent of burning. The
detection signals of the burned area were probably too low to detect the burned area along the outer edges of the fires in the
south and north (Fig. 6a). Nevertheless, this area agrees well with the actual burned area of 470 km$^2$ for the two major fire
outbreaks occurring in the municipalities of Pedrógão Grande and Góis, based on ground observations (CTI, 2017). The
modelled BC emissions (Fig. 6b), based on the FINN v1.6. database, range from less than 500 kg day$^{-1}$ pixel$^{-1}$ to over 2250
kg day$^{-1}$ pixel$^{-1}$. A total of about 203.5 tons of BC was emitted by this exceptional fire event (Table 3). Due to cloud
obscuration and masking on 18[th] June, this value is probably underestimated.

### 3.6 Deposition fluxes

Deposition fluxes were calculated for rBC and microscopic charcoal by multiplying the concentration of the respective
compounds by the snow accumulation corrected for density. Even if these values remain uncertain due to dating limitations
of the snowpit and to the lack of detailed snowfall monitoring at JFJ, preventing us from knowing the exact duration of a
snowfall event, they constitute the first step towards a quantitative transfer function. In the case of BC, mainly wet-
deposited, another difficulty is the high dependency on the time of the precipitation event. Here, the highest atmospheric
concentrations on 22[nd] June were not archived since the precipitation event in the form of snow took place on 25[th] June,
depositing only the remaining BC particles.

For microscopic charcoal, in the case of June 25[th] (peak day), we obtained a daily deposition flux of 12 fragments m$^{-2}$ s$^{-1}$, i.e.
37,800 fragments cm$^{-2}$ yr$^{-1}$, around 20 times more than the modelled flux (Fig. 5). Compared to yearly average fluxes from
high-alpine ice archives (Table 4), the estimated influx at JFJ during the event is exceptional and cannot be explained by the



somewhat lower altitude of the JFJ site compared to the other alpine ice-core locations. The comparison with other ice archives suggests this single outstanding event deposited as many charcoal particles per day (104 fragments cm$^{-2}$ day$^{-1}$) as during an average year in other ice archives (e.g. Brugger et al. 2018a).

For rBC, considering both June 25$^{th}$ and 26$^{th}$ and a cumulative snowfall duration of only 6 hours per day, as rBC is mainly wet-deposited and snowfalls were intermittent on those days as shown by the JFJ webcam, deposition fluxes amount to 1420

ng m$^{-2}$ s$^{-1}$. These values are two orders of magnitude higher than the modelled deposition fluxes (Fig. 5), which could be due to an underestimation of the fire emissions or result from both modelled and experimental uncertainties. Such large discrepancies between modelled and actual deposited rBC fluxes were already pointed out by Thomas et al. (2017), who found a factor of 2–100 in a case study from Greenland and advocated for a better description of precipitation scavenging and fire emissions by the models.

**Conclusions**

In this case study, biomass burning emissions from an outstanding fire event in Portugal in June 2017 were observed at the high-alpine site Jungfraujoch, Swiss Alps, in both the atmosphere and the snowpack. According to satellite observations, the fire burned a total area of 501 km$^2$ from 17$^{th}$ to 24$^{th}$ June, in close agreement with ground observations. At least about 203.5 tons of BC were emitted during this event. Atmospheric backward trajectory analyses showed that the resulting plume of

smoke traveled three days before reaching Switzerland, leading to a peak in atmospheric eBC at JFJ on 22$^{nd}$ June, lasting until 25$^{th}$ June when snowfall occurred, therefore archiving rBC and microscopic charcoal in the snowpack. Outstanding concentrations and influx were observed for microscopic charcoal. Our study highlights that, for microscopic charcoal, snow and ice archives are more sensitive to distant events than sedimentary archives, due to the special settings at high elevation. For snow and ice archives, it also reveals that a much longer traveling distance (≈1500 km) than previously thought can be

reached, with outstandingly high concentrations in the case of events with optimal climate and transport conditions, thus making microscopic charcoal an excellent biomass burning tracer in ice archives. On the contrary, rBC concentrations were not exceptionally high. rBC seemed to be predominantly scavenged by wet deposition, with scavenging ratios of 81–91, in line with previous studies. Simulations with a global aerosol climate model supported that the observed microscopic charcoal particles probably originated from the fires in Portugal. For BC, the model did not reproduce the observed signal due to the

predominance of other emission sources, while for charcoal, a better agreement was observed as charcoal does not have other emission sources than fires. Such case studies are important for future ice-core studies, as they document how biomass burning information is preserved in snow archives. Nevertheless, an exhaustive quantification of the process remains challenging due to the intrinsic uncertainties of each parameter, which requires further collaboration between the different disciplines involved.



## Author contributions

DO designed the project, carried out sampling, performed rBC analyses and wrote the paper. SB performed microscopic charcoal analyses and contributed to the manuscript writing. AG made atmospheric transport simulations and contributed to the manuscript writing. HW retrieved and analyzed satellite data and contributed to the manuscript writing. MiS organized and led the snowpit study, and commented the manuscript. RLM provided the eBC data from JFJ and advice to interpret them, and commented the manuscript. CS, WT and SW contributed to the manuscript writing. MaS led the project, commented and supervised writing of the manuscript.

## Competing interests

The authors declare that they have no conflict of interest.

## Acknowledgements

We thank the Swiss National Science Foundation (SNF) for granting the Sinergia project "Paleo fires from high-alpine ice cores", which funded this research (CRSII2_154450). Furthermore, we thank Nicolas Bukowiecki for providing the atmospheric eBC data from Jungfraujoch, the Institute for Snow and Avalanche Research (SLF) for the precipitation data and Sabina Brütsch for ion chromatography analyses. The online eBC measurements at Jungfraujoch were conducted with financial support from MeteoSwiss (GAW-CH aerosol monitoring program) and from the European Union as well as the Swiss State Secretariat for Education, Research and Innovation (SERI) for the European Research Infrastructure for the observation of Aerosol, Clouds and Trace Gases (ACTRIS). The International Foundation High Altitude Research Station Jungfraujoch and Gornergrat (HSFJG) is acknowledged for hosting the online aerosol measurements and for giving access for snow sampling. The authors gratefully acknowledge the personal communication and data provided by C. Wiedinmyer, Cooperative Institute for Research and Environmental Sciences (CIRES), University of Colorado Boulder, Boulder, USA. The modelling was supported by a grant from the Swiss National Supercomputing Centre (CSCS) under project ID s652. The ECHAM-HAMMOZ model is developed by a consortium composed of ETH Zürich, Max Planck Institut für Meteorologie, Forschungszentrum Jülich, the University of Oxford, the Finnish Meteorological Institute, and the Leibniz Institute for Tropospheric Research, and managed by the Center for Climate Systems Modeling (C2SM) at ETH Zürich. The MOD14A1/MYD14A1 and MCD64A1 data products were retrieved from the online Data Pool, courtesy of the NASA Land Processes Distributed Active Archive Center (LP DAAC), USGS/Earth Resources Observation and Science (EROS) Center, Sioux Falls, South Dakota, https://lpdaac.usgs.gov/data_access/data_pool.



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



**Table 1: Values used for the calculation of rBC scavenging ratios (W) at Jungfraujoch, Switzerland. For air concentration ($C_a$), the eBC average value of June 24th was used, under two different MACs. For snow concentration ($C_s$), concentrations from the two sets of replicates were used.**

| MAC (m$^2$ g$^{-1}$) | $C_a$ (ng m$^{-3}$) | $C_s$ (ng g$^{-1}$) | W |
|---|---|---|---|
| 10 | 147.5 | 7.14–8.01 | 41–46 |
| 20 | 73.8 | 7.14–8.01 | 81–91 |


**Table 2: Examples of BC scavenging ratios available in the literature.**

| Location | W | BC measurement quantity | Authors |
|---|---|---|---|
| Jungfraujoch (JFJ) | 81–91 | rBC/eBC | This study |
| Arctic | 160 | eBC | Clarke and Noone, 1985 |
| Abisko, Sweden | 97 ± 34 | eBC | Noone and Clarke, 1988 |
| Antarctica | 150 | eBC | Warren and Clarke, 1990 |
| N-E China | 140 ± 100 | EC | Wang et al., 2014 |
| Svalbard | 98 ± 46 | eBC | Gogoi et al., 2016 |
| Antarctic | 120 ± 23 | eBC | Gogoi et al., 2018 |
| Global | 125 | Modelled | Jacobson, 2004 |

**Table 3: Burned area and BC emissions per day for the June 2017 forest fire in Portugal (*cloud covered).**

| Day | Burned area (km$^2$) | BC emissions (tons) |
|---|---|---|
| 17th June | 13 | - |
| 18th June | 148* | 53.7 |
| 19th June | 243 | 99.8 |
| 20th June | 45 | 25.5 |
| 21st June | 30 | 24.5 |
| 22nd June | 19 | - |
| 23rd June | 3 | - |
| Total | 501 | 203.5 |




**Table 4: Comparison of yearly microscopic charcoal influx at Jungfraujoch with selected glacier sites based on identical laboratory preparation and analytical methods.**

| Site | Altitude (m asl) | Microscopic charcoal influx (fragments $cm^{-2}$ $year^{-1}$) | Reference |
|---|---|---|---|
| Jungfraujoch | 3560 | 37800 | This study |
| Tsambagarav (Mongolian Altai) | 4100 | 200 | Brugger et al. (2018a) |
| Colle Gnifetti (Swiss Alps) | 4500 | 390 | Gilgen et al. (2018) |
| Summit (Central Greenland) | 3200 | 9 | Brugger et al. (2019b) |
| Illimani (Bolivian Andes) | 6300 | 130 | Brugger et al. (2019a) |



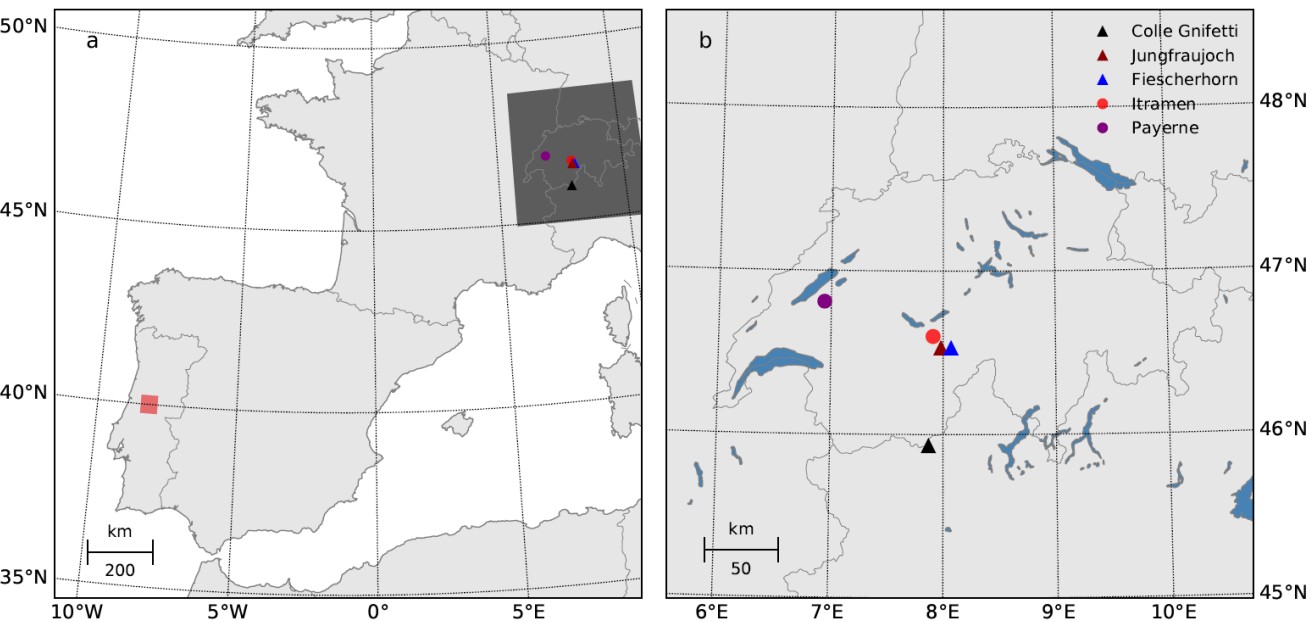

**Figure 1: Source and deposition sites. a) Map of South-Western Europe with the area of Pedrógão Grande, in Central Portugal, where the fires burned (red box) and the area of the zoom in on the right panel (grey box). b) Map of Switzerland with the sites of interest mentioned in the study. Triangles indicate high-altitude ice-core and snow study sites, circles stand for weather stations.**






**Figure 2: a) Atmospheric equivalent black carbon (eBC) concentrations at Jungfraujoch showing the peak on 22$^{nd}$ June when the plume of smoke reached the site. A mass absorption coefficient of 10 m$^2$ g$^{-1}$ was assumed. Blue bars indicate days with significant snowfall at JFJ, with a comparison between the daily precipitation amount measured at Itramen (values in blue, in mm, data from SLF © 2019, SLF) and the daily snowfall height inferred from the snowpit at JFJ, corrected for density (values in orange, in mm water equivalent (weq)). b) Snowpit stratigraphy with ice lenses in dark blue. c) Density profile of the snowpit. d) rBC concentration profile (top scale) with the two series of replicates (black and grey) and sample number associated. Mode of the rBC mass size distribution (bottom scale) with the two series of replicates (dark red and red). Blue bars are ice lenses. e) Microscopic charcoal concentration record.**



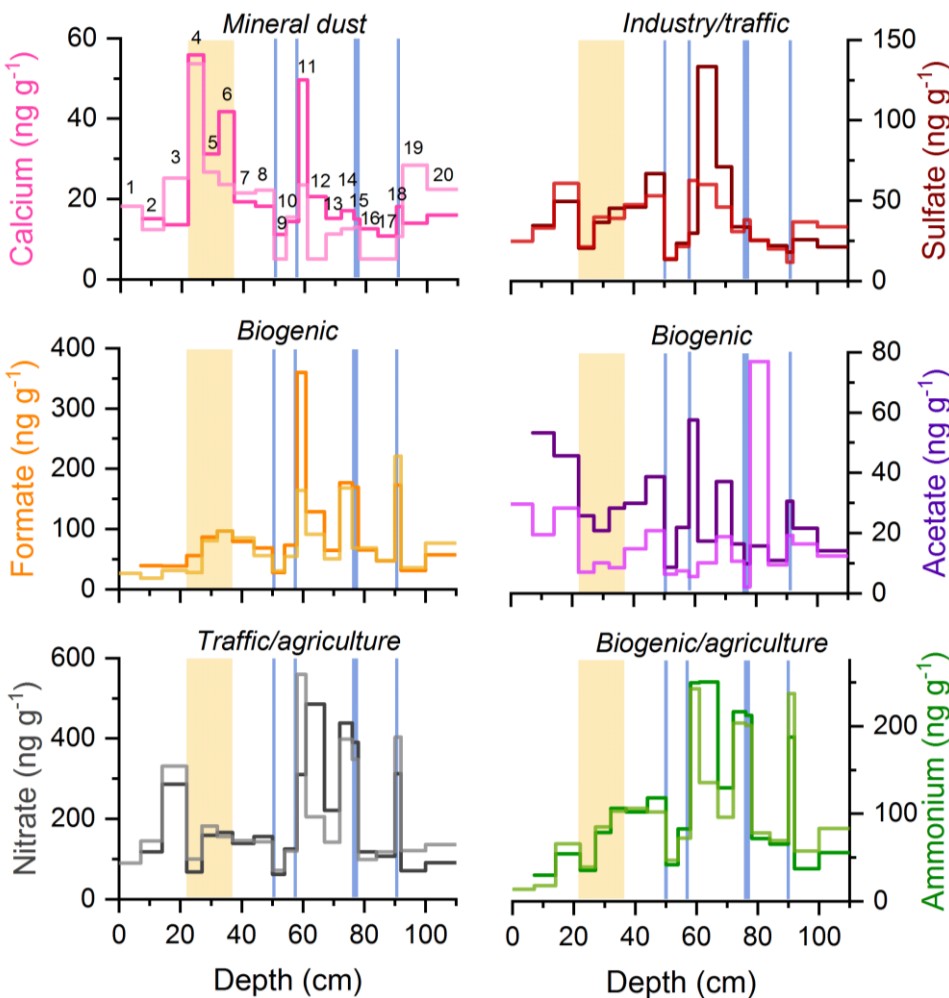

**Figure 3: Ionic records from the Jungfraujoch snowpit with the two sets of replicates (darker/lighter lines). The orange bar indicates the depth at which rBC and microscopic charcoal peaks are observed. Blue bars represent ice layers. Sample numbers are specified for calcium and are similar for the other ions. Potential sources are indicated above each graph.**



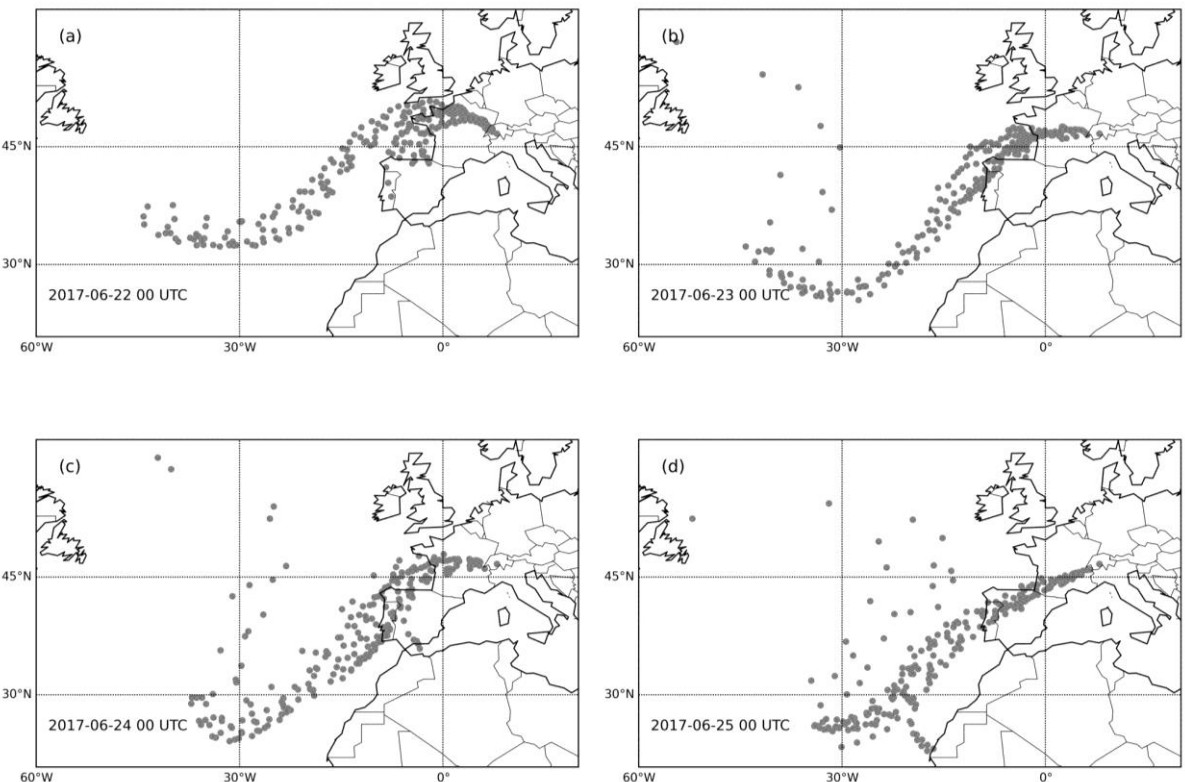

**Figure 4: 3-day air mass backward trajectories starting at the Jungfraujoch site for the 22$^{nd}$ to the 25$^{th}$ June 2017.**



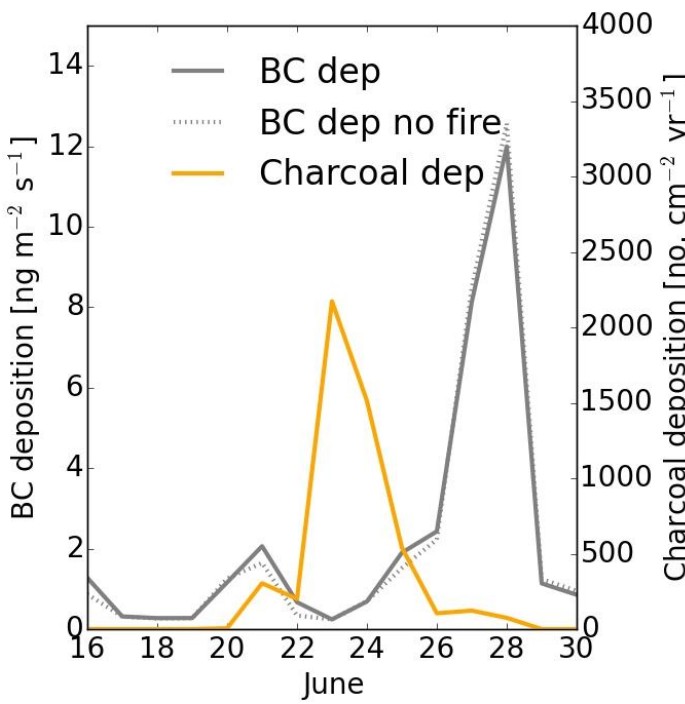

**Figure 5: Simulated deposition fluxes of BC and microscopic charcoal at Jungfraujoch. In line with the observations, the deposition (dep) fluxes for BC are given as mass fluxes, whereas the deposition fluxes of microscopic charcoal are given as the number fluxes of particles larger than a certain threshold (10 µm major axis). For BC, a simulation without fire emissions (dotted gray line) and a simulation including the fire emissions near Pedrógão Grande in Central Portugal (solid gray line) are shown. For microscopic charcoal, only the simulation including fire emissions (solid orange line) is shown since biomass burning is the only source of microscopic charcoal.**



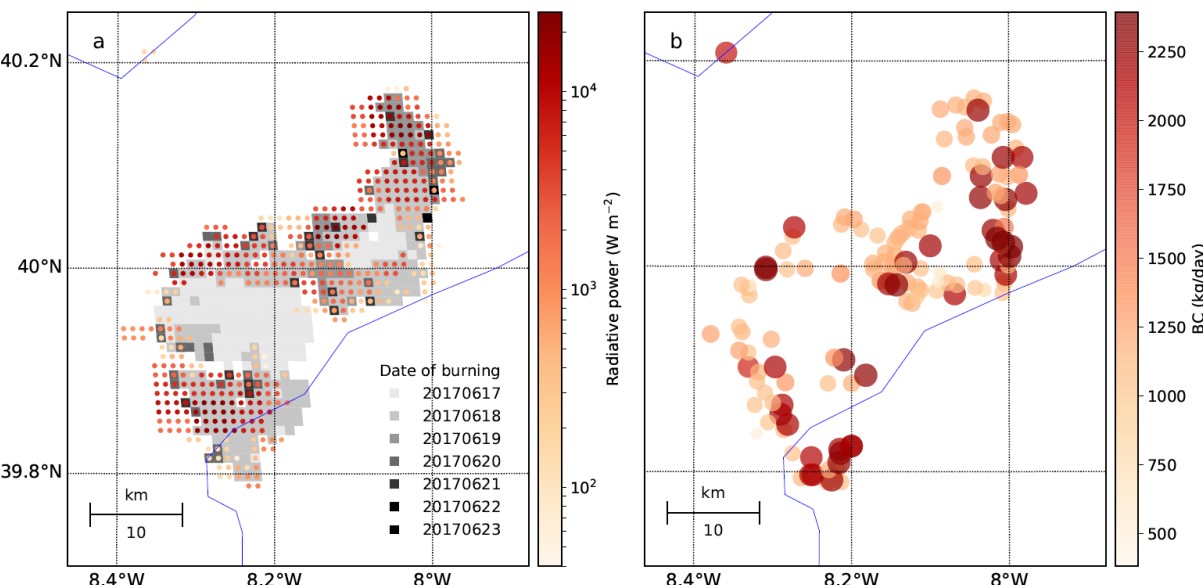

**Figure 6: Spatial and temporal observed wild fires and their BC emissions near Pedrógão Grande, Portugal, for the 17th to 23th of June at 1km spatial resolution. a) Date of the area burned (MCD64A1) and maximum Fire Radiative Power (FRP) values of the active fires according to MOD14A1/MYD14A1 showing the evolved two fire clusters. b) Corresponding BC emission values based on the FINN v1.6 database with high values at the outer edges.**