# Peer review of "Tracing devastating fires in Portugal to a snow archive in the Swiss Alps: a case study"

_The Cryosphere, 2020_

## Short Comment (SC1) · 6 Mar 2020

This is interesting work. Just a short comment in relation to the burned area estimate. The fire blowup, with extremely high rate of spread and pyCb formation and collapse that killed 66 people, occurred on the first day of its development. As such your 17th June burned area estimates are underestimated by one order of magnitude. According to our reconstruction (CTI 2017), resulting from the combination of various ground- and remote sensing-based information, burned area on the 1st and 2nd days was 128 and 211 km2, respectively. Thus remote sensing products did not detect fire growth nor peak FRP happening at about 19-21 h PM on the 17th, presumably because of the combination of dense smoke with thunderstorm clouds.

---

## Referee Comment (RC1) · Anonymous Referee #1 · 11 Mar 2020

This manuscript describes a very interesting study attempting to make quantitative connection between emissions of BC, charcoal, and ionic species by fires burning in Portugal from 17 to 24 June 2017 and their deposition to snow near Jungfraujoch (JFJ) in the Bernese Alps. The case for charcoal is compelling, while the findings for BC and ionic tracers of smoke are not so clear.

A combination of remote sensing and in-situ atmospheric sampling, plus back trajectory and chemical transport modeling do show that smoke from the subject fires was transported to JFJ and was observed in the atmosphere above the snowpit site from 22 through 24 June. Atmospheric concentrations of BC dropped sharply on 25 June, coincident with significant snowfall at JFJ, and remained low until the end of June. Detailed

stratigraphy and sampling for chemical analysis in a 1 m deep snow pit showed that layers representing 3 different snow events (on 25-26, 28, and 29 June) were present in the top 40 cm of the pit. Concentrations of BC were modestly enhanced and the abundance of charcoal fragments hugely increased in the layer from the first snowfall (25-26 June), but none of the ions often suggested to be smoke tracers (formate, ammonium, potassium, acetate, nitrate) were elevated. It should be noted that BC and the ions were measured in the same samples that were nominally 5 cm depth resolution while charcoal fragments were quantified in samples collected at 10 cm resolution; the BC signal in the pit appeared in 3 samples between $\sim$22 and 38 cm depth while enhanced charcoal was in a single sample covering the 30 to 40 cm depth range. It is unfortunate that the different records are not all at the same depth resolution, but the fact that the sample with peak charcoal contained some fraction of "older" snow than the deepest of the samples with elevated BC becomes important when the model results are considered. It is not possible to say from the information provided in the manuscript whether the snow between 38 and 40 cm just fell early in the event on 25 June, or included snow that had fallen days earlier, but the model suggests that much of the charcoal deposition occurred during 23-24 June compared to peak deposition of BC on the 27th and 28th.

The model also suggests that the BC peak is not due to smoke, rather it just reflects efficient scavenging of regional pollution by snow falling mainly on 27 and 28 June. The pit stratigraphy suggests that most of the BC is in snow that fell on 25-26 June, but there is some ambiguity in depth to age conversions. The authors suggest that coarse spatial resolution of the model prevents it from accurately capturing the fire emitted BC on top of a large regional background. That may be partially true as suggested by the fact that it did not actually snow at JFJ on the 27th and only a small amount of rain was observed at the nearest weather station. However I find the performance of the model to be surprisingly good, and urge the authors to consider the possibility that while the charcoal is doubtless dominated by smoke from Portugal, the BC may be essentially just a mix of European pollution. I will first outline evidence that the model is closely

reproducing the observed deposition of both BC and charcoal, and then suggest some ways the authors might be able to convince me, and other readers, that the BC is actually dominated by deposition from the smoke plume.

In section 3.6 the authors suggest that the CTM underestimates charcoal deposition by about a factor of 20 and that of BC by two orders of magnitude. It is not entirely clear how the fluxes were estimated from the observations in the snow pit since there is some ambiguity regarding the proper timescale, but here is a straightforward approach that suggests much better agreement.

The charcoal sample # 2 contains 20,000 fragments/L of snow or 20 fragments/g. The density of 0.54 g snow/cm^3 times the depth of the sample indicates that the 30-40 cm layer contains 5.4 g snow/cm^2, when multiplied by 20 fragments/g this indicates that 108 fragments/cm^2 were deposited (total for the event, not per second, day or year). From Fig 5 I estimate that the model deposited this on 23, 24, and 25 June at rates of 2200, 1500, and 600 fragments/cm^2 y, respectively. Using simple average of 14,333 fragments/cm^2 y x 1/365 d/year x 3 days gives modeled deposition of 118 fragments/cm^2 which is almost too close to 108 to be possible, given concerns about the model and especially the emissions of charcoal by the fire (scaled to BC estimates from a completely different model).

Similar calculation for BC starts with estimate of 7.5 ng/g (average of the 6 samples #s 4-6, all replicated, in Fig 2) x 0.54 g/cm^3 x 15 cm (depth of the 3 samples combined) yielding total burden of 61 ng BC/cm^2 in this layer. For the model estimate an eyeball average of the calculated flux over 26-28 June is 6 ng/m^2 sec x 3 d x 86,400 sec/d x 1/10,000 cm^2/m^2 = 155 ng BC/cm^2. Not as close as the agreement between observed and modeled charcoal deposition but a factor of 2.5 is nowhere near 100-fold difference.

To me, this suggests that the scenario suggested by the model is plausible even if not precisely correct in detail. Passage of smoke over JFJ caused dry deposition of charcoal sometime (hours or maybe a few days) before it started snowing on 25 June. Very little BC or ionic smoke tracers were removed by this process. Then a change in transport just before or coincident with the snow fall event brought air with regional pollution but very little or no smoke from Portugal to JFJ. Wet deposition via the snow created an anthropogenic BC enhancement that lacks any formate, ammonium, potassium, etc.

Using a regional CTM with grid size in the 4-10 km range rather than the global version selected initially might help to clarify whether the BC is linked to the fires rather than being mostly regional pollution. A simpler/cheaper, but also complementary, approach would be to run forward trajectories from the fire, possibly over the entire 17-24 June lifetime but at least beginning early enough to capture the first time smoke reaches JFJ on 22 June and continuing until the fire is out. In the scenario laid out in the manuscript these trajectories would have to show strong connection between the fire and JFJ lasting well into 25 June, while the alternative outlined above predicts that the smoke clears out over JFJ before it starts snowing.

As noted right at the beginning, this is an interesting story, and the firm results for charcoal make it important to get before the community. I think that the argument linking the BC in the snow to the fires needs to be made much more convincingly, or ruled out just as strongly. Neither option would impact the charcoal connection, while insisting that the BC is fire derived based on weak evidence lessens the power of the manuscript.

Following are a list of specific comments and editorial suggestions, keyed to line number.

27 As noted above, the correspondence between eBc measured through 24 June and rBc measured in snow that fell 25-26 June may be more tenuous than asserted.

28-29 Calculated scavenging ratios may be oversold since there is no assurance that BC at cloud height 25 and 26 June was same as inferred from measured eBc on the ground 24 June.

33-34 "This study unambiguously links charcoal in the snow with the highly intensive fires in Portugal..." At least one reader is not convinced that rBC in the snow is from these fires, and would liked to have seen some of the ionic tracers supporting that inference yet none do.

35 Is the BC emission estimate not basically straight from FINN, rather than ECHAM?

39 what do you mean by "landscape" fires, as something distinct from biomass fires

50 consider citing some of the pioneering studies of fire tracers in polar ice cores, for example; Legrand et al., 1992 (GRL); Whitlow et al., 1994 (Tellus; Legrand and De Angelis, 1996 (JGR)

92-94 Sentence pointing out that ice cores from near JFJ have been studied is not needed unless you later make some connection to the cited papers.

97 located on the eponymous pass between—→ located between

135-138 Might want to state the dates for which backtrajectories were calculated. And as noted above, consider running forward trajectories from the fire as well.

139-149 If you cannot, or decide not to, run a regional model, I think you should provide more justification for choosing to run this particular version of ECHAM, especially since you kind of denigrate its performance later in the manuscript 151-157 Curious why you chose to only use MODIS products. It is becoming increasingly clear that important details are missed due to coarse spatial resolution, and the fixed single overpass time. Are there not relevant products from the Sentinel satellites? Geostationary platforms (mainly supporting meteorology forecasting) can provide insight throughout the day, especially later in the afternoon when many fires are strongest.

184 I would not say that peak rBC of 9.8 ng/g is "remarkable". It is only about 2 x higher than the secondary peak at 60-70 cm depth. The Thomas et al., 2017 paper cited elsewhere found the average peak in 22 north Greenland pits to be 15 ng/g, with max of 43 ng/g with longer transport distances back to the source fires.

193-194 I would mention the secondary rBC peak in samples 12/13 that overlaps the bump up in charcoal sample 5, especially since you point out the increased size later to suggest more local source.

197 Regarding the "narrower" charcoal peak, I think it may mostly be in the 2 cm interval 28-30 cm below rBC sample 6, but that is just a hypothesis.

270-280 As noted above, consider forward trajectories from the fires, specifically looking to see if smoke was likely over JFJ when it started snowing 25 June. And seriously consider whether ECHAM is possibly correct that the BC in the snow is not from the fires.

Section 3.5 See earlier question about exclusive reliance on MODIS, and the on-line comment from Paolo Fernandes.

Section 3.6 Consider the "deposited" BC and charcoal calculations presented in introductory comments. If you decide to stick with flux estimates provide more details about assumptions used to get values so much higher than the model.

Last paragraph of this section, seems that the model thought there was no smoke at all over JFJ, pointing to a transport shift (or error) rather than problems with emissions. I made a case that the model came within factor of 2.5 of the total amount of BC in the snow from 25-26 June, even though it wants to deposit most of it on the 27th and 28th . Might be worth comparing the timing and amount of precipitation in the model to observations.

---

## Referee Comment (RC2) · Anonymous Referee #2 · 6 Apr 2020

This manuscript attempts to link emissions from a known fire event and their deposition on snow close to JFJ in the Swiss Alps through extensively investigating a severe fire event occurred on 17-24 June 2017 in Portugal. This study is attempting to provide an interesting approach, connecting a set of valuable record of charcoal, black carbon, and ions in the snow pit and a combination of atmospheric in-situ measurement, remote sensing, air mass trajectory calculation, and transport simulation, to better understand the information of particle deposition in snow in European high-altitude sites. This approach is very useful to understand atmospheric processes of aerosol particles how to transport over long distances, be scavenged by snow fall, and be deposited to snow by providing clues. However, there is an issue that should be considered seriously to publish the results. Authors measured charcoal, refractory black carbon (rBC), and ions

in snow collected near JFJ and compared their profiles with equivalent black carbon (eBC) measured in the atmosphere at JFJ. The elevations of charcoal in the upper layer of C in snow (sample "2" in figure 2e) and eBC in atmosphere on 22 June (figure 2a) were obvious with an increase relative to background level by ~6 times and ~10 times, respectively, indicating that the fire plume reached JFJ. In contrast to them, rBC in sample "4-6" (figure 2d) increased a little and changes in ion concentrations are even indiscernible. Readers reasonably expect that potassium, ammonium, and nitrate can be elevated in their concentrations if the big fire plume indeed reached and is detectable by charcoal and eBC. In the same snow layer, the rBC concentration was elevated just 1.5 times relative to the second peak with rBC concentration of ~5 ng/g, that is probably from local fires as suggested by authors. Discuss this issue in depth and provide more evidences and/or assumption to support your argument.

L135: Since the fire event studied in this work was a well-documented recent extreme event, authors need to calculate forward air mass trajectories to see if the plume departing from the exact fire spot reach the JFJ site. Consider that a starting height of air mass trajectories is not necessarily just above ground because fire-emitted particles can be directly injected into the free troposphere up to > 3 km. Also, it would be great to see if the area-averaged time series of AOD match with forward air mass trajectories. For AOD, just simply check GIOVANNI platform (https://giovanni.sci.gsfc.nasa.gov/giovanni/#service=DiArAvTs&starttime=&endtime=&bbox=-180,-90,180,90). Intense fire plumes are often easily captured by AOD, and thus AOD near JFJ should be elevated day-by day.

L206: The upper size limit of BBHG is generally <300 nm, although which depends on the instrumental setting. Please check if authors mentioned "BBLG" instead of "BBHG".

L205-214: In figure 2d, the diameter of rBC particles in sample "4-6" (figure 2d) seem similar to that of other layers except for sample "13". The size of atmospheric rBC particles generated from biomass combustion is generally larger than that from urban fossil emissions. If the rBC size is not obviously large in sample "4-6", the possible

reason should be suggested, regarding for example, cloud or wet-scavenging during transport and/or scavenging by snowfall, etc.

L224: Authors mentioned that smoke particles can be lifted up to free troposphere and travel over long distance. It can be true not only for charcoal but also for rBC particles. As noted above, it should be seriously considered why rBC and ionic particles did not elevated unlike charcoal. rBC particles are small with diameters < 1 micrometer, which thus can travel longer distances, as found in previous studies so far. Also, authors may need to check if the number of particles in SP2 scattering channel ("SCLG" or "SCHG") is elevated in the snow layers. The profile of scattering particles, i.e., number concentration, might correspond to that of charcoal.

L230: Both ammonia (NH3) and NOx can be emitted from fires, and during the transport time in the atmosphere, ammonium nitrate (NH4NO3) and other forms of nitrate and ammonium can be formed particularly under high relative humidity via aqueous reaction. Authors should seriously consider why such inorganic ions are not sufficiently detected in the snow layer corresponding to smoke plume. Meteorological conditions, for example too dry condition in FT and/or too warm in PBL, were not favorable for the aerosol formation? Moreover, potassium has been broadly used as an indicator of biomass combustion so far. Did authors observe a peak of potassium in the snow layer?
* * *

---

## Author Comment (AC2) · 3 Jul 2020

Author's response to comment from Paulo Fernandes on

"Tracing devastating fires in Portugal to a snow archive in the Swiss Alps: a case study", by Dimitri Osmont et al., submitted to TC

We would like to thank Paulo Fernandes for his constructive comment which helped us to improve the quality of this paper. Please find below our response to your comment (in blue) and our changes to the manuscript (in grey and italic).

**Paulo Fernandes**

pamfernand@gmail.com

This is interesting work. Just a short comment in relation to the burned area estimate. The fire blowup, with extremely high rate of spread and pyCb formation and collapse that killed 66 people, occurred on the first day of its development. As such your 17[th] June burned area estimates are underestimated by one order of magnitude. According to our reconstruction (CTI 2017), resulting from the combination of various ground- and remote sensing-based information, burned area on the 1st and 2nd days was 128 and 211 km2, respectively. Thus remote sensing products did not detect fire growth nor peak FRP happening at about 19-21 h PM on the 17th, presumably because of the combination of dense smoke with thunderstorm clouds.

Thank you for your comment. We actually analysed remote sensing data for the 17[th], but did not describe this in detail in the manuscript. On the 17[th], the fire was early detected by Soumi/VIIRS (375m) about 13:46-13:47 UTC few kilometres from a thick cumulonimbus cloud. For Aqua/MODIS 1km data (overpass time 13:45-13:50 UTC – almost simultaneous overflight), the fire was too small to be detected. This explains why FINN has no emission entries for the 17[th]. This was included in the manuscript.

*An additional comparison of the Aqua/MODIS and VIIRS/NPP Active Fire Product (375 m spatial resolution; Schroeder at el., 2014) showed, that the fire was early detected by the VIIRS/NPP Fire Product at ~13:46 UTC. The fire was located few kilometres away from a thick cumulonimbus cloud. Though, the overpass time of Aqua/MODIS was almost simultaneous (i.e. 13:45-13:50 UTC), the fire was too small to be detected by the coarser spatial resolution sensor (1 km). This explains why FINN v1.6 has no emission entries for 17[th] June.*

---

## Author Response (AR2)

**Author's response to referee comments on**

"Tracing devastating fires in Portugal to a snow archive in the Swiss Alps: a case study", by Dimitri Osmont et al., submitted to TC

We would like to thank the referees, Paulo Fernandes and the editor for the time spent on our manuscript

5  and for the detailed and constructive comments which helped us to improve the quality of this paper. Please find below our responses to their comments (in blue) and our changes to the manuscript (in grey and italic).

**Anonymous Referee #1**

10  Osmont et al.

This manuscript describes a very interesting study attempting to make quantitative connection between emissions of BC, charcoal, and ionic species by fires burning in Portugal from 17 to 24 June 2017 and their deposition to snow near Jungfraujoch (JFJ) in the Bernese Alps. The case for charcoal is compelling, while the findings for BC and ionic tracers of smoke are not so clear.

15  A combination of remote sensing and in-situ atmospheric sampling, plus back trajectory and chemical transport modeling do show that smoke from the subject fires was transported to JFJ and was observed in the atmosphere above the snowpit site from 22 through 24 June. Atmospheric concentrations of BC dropped sharply on 25 June, coincident with significant snowfall at JFJ, and remained low until the end of June. Detailed stratigraphy and sampling for chemical analysis in a 1 m deep snow pit showed that

20  layers representing 3 different snow events (on 25-26, 28, and 29 June) were present in the top 40 cm of the pit. Concentrations of BC were modestly enhanced and the abundance of charcoal fragments hugely increased in the layer from the first snowfall (25-26 June), but none of the ions often suggested to be smoke tracers (formate, ammonium, potassium, acetate, nitrate) were elevated. It should be noted that BC and the ions were measured in the same samples that were nominally 5 cm depth resolution while charcoal

25  fragments were quantified in samples collected at 10 cm resolution; the BC signal in the pit appeared in 3 samples between _22 and 38 cm depth while enhanced charcoal was in a single sample covering the 30 to 40 cm depth range. It is unfortunate that the different records are not all at the same depth resolution, but the fact that the sample with peak charcoal contained some fraction of "older" snow than the deepest of the samples with elevated BC becomes important when the model results are considered. It is not

30  possible to say from the information provided in the manuscript whether the snow between 38 and 40 cm just fell early in the event on 25 June, or included snow that had fallen days earlier, but the model suggests that much of the charcoal deposition occurred during 23-24 June compared to peak deposition of BC on the 27th and 28th.

The model also suggests that the BC peak is not due to smoke, rather it just reflects efficient scavenging

35  of regional pollution by snow falling mainly on 27 and 28 June. The pit stratigraphy suggests that most of the BC is in snow that fell on 25-26 June, but there is some ambiguity in depth to age conversions. The

authors suggest that coarse spatial resolution of the model prevents it from accurately capturing the fire emitted BC on top of a large regional background. That may be partially true as suggested by the fact that it did not actually snow at JFJ on the 27th and only a small amount of rain was observed at the nearest weather station. However I find the performance of the model to be surprisingly good, and urge the authors to consider the possibility that while the charcoal is doubtless dominated by smoke from Portugal, the BC may be essentially just a mix of European pollution. I will first outline evidence that the model is closely reproducing the observed deposition of both BC and charcoal, and then suggest some ways the authors might be able to convince me, and other readers, that the BC is actually dominated by deposition from the smoke plume.

In section 3.6 the authors suggest that the CTM underestimates charcoal deposition by about a factor of 20 and that of BC by two orders of magnitude. It is not entirely clear how the fluxes were estimated from the observations in the snow pit since there is some ambiguity regarding the proper timescale, but here is a straightforward approach that suggests much better agreement.

The charcoal sample # 2 contains 20,000 fragments/L of snow or 20 fragments/g. The density of 0.54 g snow/cm^3 times the depth of the sample indicates that the 30-40 cm layer contains 5.4 g snow/cm^2, when multiplied by 20 fragments/g this indicates that 108 fragments/cm^2 were deposited (total for the event, not per second, day or year). From Fig 5 I estimate that the model deposited this on 23, 24, and 25 June at rates of 2200, 1500, and 600 fragments/cm^2 y, respectively. Using simple average of 14,333 fragments/cm^2 y x 1/365 d/year x 3 days gives modeled deposition of 118 fragments/cm^2 which is almost too close to 108 to be possible, given concerns about the model and especially the emissions of charcoal by the fire (scaled to BC estimates from a completely different model).

We appreciate the careful reconsidering of the fluxes and agree with the reviewer that the approach of calculating total fluxes is more straightforward, circumventing the uncertainty in the time scales. However, the reviewer used a wrong average value for the charcoal concentrations (14333 instead of 1433). We get an integral charcoal flux of 13.8 fragments cm$^{-2}$ from the model, compared to a measured total charcoal flux of 104 fragments cm$^{-2}$, which is still a factor of 8 underestimation by the model.

Similar calculation for BC starts with estimate of 7.5 ng/g (average of the 6 samples #s 4-6, all replicated, in Fig 2) x 0.54 g/cm^3 x 15 cm (depth of the 3 samples combined) yielding total burden of 61 ng BC/cm^2 in this layer. For the model estimate an eyeball average of the calculated flux over 26-28 June is 6 ng/m^2 sec x 3 d x 86,400 sec/d x 1/10,000 cm^2/m^2 = 155 ng BC/cm^2. Not as close as the agreement between observed and modeled charcoal deposition but a factor of 2.5 is nowhere near 100-fold difference.

The measured total BC flux of 61 ng cm$^{-2}$ is correct and the modelled integral flux is 195 ng cm$^{-2}$, so the model overestimates the flux by a factor of 3. Interestingly, the model underestimates the charcoal and overestimates the BC flux. When the model performance was evaluated with longer-term charcoal deposition fluxes, the opposite was observed (Gilgen et al., 2018). It was hypothesized that the model overestimates the fluxes at ice core sites because of their high location within complex topography. The model is not able to simulate these high locations correctly since the surface altitude is constant over the whole grid box; i.e. the topography is smoothed. In addition, ice cores are often located above the top plume height of most fires (Rémy et al., 2017), which may prevent transport of charcoal particles to them. Obviously the Pedrógão Grande case was exceptional, since the plume was transported at elevations

between 3000 and 5000 m a.s.l.. In addition, an underestimation of the fire emissions might have played a role. In contrast, the smoothed topography in the model and the corresponding more efficient vertical mixing might have resulted in overestimated levels of BC from regional anthropogenic sources, explaining the overestimation of the BC flux. We revised the manuscript accordingly:

*L323-328: For microscopic charcoal, we observed a total deposition flux of 104 fragments cm$^{-2}$ in the snowpit, around 8 times more than the modelled flux of 13.8 fragments cm$^{-2}$ (integral over 23$^{rd}$ and 24$^{th}$ June, Fig. 5). Compared to yearly average fluxes from high-alpine ice archives (Table 4), the estimated influx at JFJ during the event is exceptional and cannot be explained by the somewhat lower altitude of the JFJ site compared to the other alpine ice-core locations. The comparison with other ice archives suggests this single outstanding event deposited nearly as many charcoal particles as during an average year in other ice archives (e.g. Brugger et al. 2018a).*

*L329-342: For rBC, the total deposition flux for samples 4, 5, and 6 was 62 ng cm$^{-2}$ (average of replicate samples). The integral flux retrieved from the model for 26$^{th}$, 27$^{th}$, and 28$^{th}$ June is 195 ng cm$^{-2}$ (Fig. 5); a factor of three higher than the observation. Interestingly, the model underestimates the charcoal and overestimates the BC flux. When the model performance was evaluated with longer-term charcoal deposition fluxes, the opposite was observed (Gilgen et al., 2018). It was hypothesized that the model overestimates the fluxes at ice core sites because of their high location within complex topography. The model is not able to simulate these high locations correctly, since the surface altitude is constant over the whole grid box; i.e. the topography is smoothed. In addition, ice core sites are often located above the top plume height of most fires (Rémy et al., 2017), which may prevent transport of charcoal particles to them. Obviously the Pedrógão Grande case was exceptional, since the plume was transported at elevations between 3000 and 5000 m a.s.l.. In addition, an underestimation of the fire emissions might have played a role for the charcoal flux. Contrary to the effect on charcoal, the smoothed topography in the model and the corresponding more efficient vertical mixing might have resulted in overestimated levels of BC from regional anthropogenic sources, explaining the overestimation of the BC flux.*

To me, this suggests that the scenario suggested by the model is plausible even if not precisely correct in detail. Passage of smoke over JFJ caused dry deposition of char-coal sometime (hours or maybe a few days) before it started snowing on 25 June. Very little BC or ionic smoke tracers were removed by this process. Then a change in transport just before or coincident with the snow fall event brought air with regional pollution but very little or no smoke from Portugal to JFJ. Wet deposition via the snow created an anthropogenic BC enhancement that lacks any formate, ammonium, potassium, etc.

This is a really helpful suggestion, and the point was raised also by referee 2. We were from the beginning puzzled by not finding elevated concentrations of the other fire tracers (ammonium, formate, and acetate) in the samples 4-6. Re-considering this in view of both referee's comments, we revised our interpretation. Charcoal originating from the Portugal fires was deposited by dry deposition during the period 22$^{nd}$-24$^{th}$ June, when the fire plume was detected by atmospheric eBC measurements at JFJ. Since no snowfall occurred during that period, the majority of BC was not deposited, since dry deposition is not efficient for submicron particles. Dry deposition of charcoal most likely resulted in a confined layer, which was not resolved with the 10 cm sampling resolution. With the snowfall on 25$^{th}$ June, air mass transport changed, ending the advection of the fire plume to the JFJ as indicated by the backward trajectories

arriving at JFJ at 18 UTC (will be included in Fig. 4 in the revised version). Instead more regional polluted air masses were scavenged. That the charcoal and rBC peaks partly overlap is most likely due to the coarse resolution of the charcoal samples. The change of air masses does explain the lack of a peak in the other fire tracers, e.g. ammonium, and the time delay between the charcoal and the BC peak in the model output and the snow pit data. The manuscript was changed accordingly.

*Abstract:*

*L26-38: According to modelled emissions of the FINN v1.6 database, the fire emitted a total amount of 203.5 tons BC from a total burned area of 501 km$^2$ as observed on the basis of satellite fire products. Backward trajectories unambiguously linked a peak of atmospheric equivalent BC observed at the Jungfraujoch research station on 22$^{nd}$ June, with elevated levels until the 25$^{th}$ June, with the highly intensive fires in Portugal. The atmospheric signal is in correspondence with an outstanding peak in microscopic charcoal observed in the snow layer, depositing nearly as many charcoal particles as during an average year in other ice archives. In contrast to charcoal, the amount of atmospheric BC deposited during the fire episode was minor due to a lack of precipitation. Simulations with a global aerosol climate model supported that the observed microscopic charcoal particles originated from the fires in Portugal and that their contribution to the BC signal in snow was negligible. Our study revealed that microscopic charcoal can be transported over long distances (1500 km), and that snow and ice archives are much more sensitive to distant events than sedimentary archives, for which the signal is dominated by local fires. The findings are important for future ice-core studies, as they document that for BC as fire tracer the signal preservation depends on precipitation. Single events, like this example, might not be preserved due to unfavorable meteorological conditions.*

*Fire tracers:*

*L237-241: We hypothesize that charcoal originating from the Portugal fires was deposited by dry deposition during the period 22$^{nd}$ to 24$^{th}$ June, when the fire plume arrived at the JFJ as detected by elevated atmospheric eBC concentrations. Since no snowfall occurred during that period, the majority of BC was not deposited. Dry deposition most likely resulted in a confined charcoal layer, which was not separated from the rBC peak in the snowpit due to the coarse sampling resolution of 10 cm. With the beginning of snowfall on 25$^{th}$ June, air mass transport changed, ending the advection of the fire plume to the JFJ as indicated by backward trajectories (see below and Fig. 4). Instead more regional polluted air masses were scavenged, which explains the absence of ionic fire tracers and of a shift in the rBC size distribution.*

*Atmospheric transport:*

*L271-274: The back trajectories with arrival at JFJ on 25$^{th}$ June at 18 UTC indicate a major change in the synoptic situation to more north-westerly flow directions. We can only speculate that this happened concomitant with the onset of precipitation, since the timing of the latter is not precisely known.*

*L279-283: Furthermore, other BC sources than fires seem to dominate in our simulations, which is in agreement with our hypothesis that during fire plume arrival at JFJ the majority of fire-related BC was not deposited due to the lack of snowfall. With the beginning of snowfall at JFJ, air mass transport changed and more regional polluted air masses with minor or without fire contribution were scavenged.*

*Conclusions:*

*L349-350: Dry deposition of microscopic charcoal resulted in an outstanding peak in the snowpack. This event deposited nearly as many charcoal particles as during an average year in other ice archives.*

*L354-363: For rBC, in contrast, concentrations in the snow were not exceptionally high. In combination with the absence of a peak in ionic fire tracers such as ammonium, this suggest that the majority of atmospheric BC was not deposited during the fire episode due to a lack of precipitation. Instead the observed rBC peak was mostly likely caused by scavenging of air masses containing regional pollution with the beginning of snowfall on 25th June, which ended the advection of the fire plume to the JFJ. rBC scavenging ratios were in line with previous studies, giving additional evidence that rBC was predominantly scavenged by wet deposition. Simulations with a global aerosol climate model supported that the observed microscopic charcoal particles originated from the fires in Portugal, whereas their contribution to the BC signal in snow was minor. The findings of our case study are important for future ice-core studies, as they document that for BC as fire tracer the signal preservation depends on precipitation and wet deposition. Single events, like this example, might not be preserved due to the unfavorable meteorological conditions.*

Using a regional CTM with grid size in the 4-10 km range rather than the global version selected initially might help to clarify whether the BC is linked to the fires rather than being mostly regional pollution. A simpler/cheaper, but also complementary, approach would be to run forward trajectories from the fire, possibly over the entire 17-24 June lifetime but at least beginning early enough to capture the first time smoke reaches JFJ on 22 June and continuing until the fire is out. In the scenario laid out in the manuscript these trajectories would have to show strong connection between the fire and JFJ lasting well into 25 June, while the alternative outlined above predicts that the smoke clears out over JFJ before it starts snowing.

We calculated forward trajectories as suggested by the reviewer and they basically agree with the backward trajectories. We prefer to keep the backward trajectories in the manuscript, since they are more conclusive, but added the one arriving at JFJ on 25th June 18 UTC to show the change to north-westerly air flow. We show now trajectories calculated 5-days backward at 20 equidistant levels in pressure coordinates between 700 hPa and 500 hPa, in accordance with the detected smoke plume height.

To the best of our knowledge, the ECHAM-HAM is the only model with a charcoal module implemented. We included that in the manuscript.

*Figure 4.*

*L142-143: To the best of our knowledge, this is the only model with a charcoal module implemented.*

As noted right at the beginning, this is an interesting story, and the firm results for charcoal make it important to get before the community. I think that the argument linking the BC in the snow to the fires needs to be made much more convincingly, or ruled out just as strongly. Neither option would impact the charcoal connection, while insisting that the BC is fire derived based on weak evidence lessens the power of the manuscript.

See comment above.

195 Following are a list of specific comments and editorial suggestions, keyed to line number.

27 As noted above, the correspondence between eBc measured through 24 June and rBc measured in snow that fell 25-26 June may be more tenuous than asserted.

Agreed, see comment above.

28-29 Calculated scavenging ratios may be oversold since there is no assurance that BC at cloud height 200 25 and 26 June was same as inferred from measured eBc on the ground 24 June.

We agree with that comment, but this is the case for most of the published scavenging ratios. Since there are so few data we decided to keep it.

33-34 "This study unambiguously links charcoal in the snow with the highly intensive fires in Portugal: : :" At least one reader is not convinced that rBC in the snow is from these fires, and would liked to have 205 seen some of the ionic tracers supporting that inference yet none do.

Agreed, see above. The sentence was deleted.

35 Is the BC emission estimate not basically straight from FINN, rather than ECHAM?

Thanks for noticing. Yes, the accumulated amount of 203.5 tons BC is based on modelled emissions on the FINN v1.6 database (see also 305). We have rewritten the sentence as in the following:

210 *L26-27: According to modelled emissions of the FINN v1.6 database, the fire emitted a total amount of 203.5 tons BC.*

39 what do you mean by "landscape" fires, as something distinct from biomass fires

Bowmann et al. (2009) distinguish between "landscape" fires, "biomass combustion for domestic and industrial uses, and fossil-fuel combustion". Thus, "including landscape and biomass" refers to landscape 215 fires and biomass combustion for domestic and industrial uses. We have clarified this accordingly.

*L41-43: Global $CO_2$ emissions from fires, including landscape and biomass (i.e. biomass combustion from domestic and industrial uses), represent around 50% of those produced by fossil fuel burning (Bowmann et al., 2009).*

50 consider citing some of the pioneering studies of fire tracers in polar ice cores, for example; Legrand 220 et al., 1992 (GRL); Whitlow et al., 1994 (Tellus; Legrand and De Angelis, 1996 (JGR)

Are now included (L53).

92-94 Sentence pointing out that ice cores from near JFJ have been studied is not needed unless you later make some connection to the cited papers.

The sentence was deleted.

225 97 located on the eponymous pass between—! located between

OK, eponymous was deleted.

135-138 Might want to state the dates for which backtrajectories were calculated. And as noted above, consider running forward trajectories from the fire as well.

Was included in the caption of Figure 4. Forward trajectories were calculated, see above.

*Figure 4: 5-day air mass backward trajectories starting from the Jungfraujoch site at 00 UTC on 22$^{nd}$ to 24$^{th}$ June 2017. For 25$^{th}$ June starting time is 18 UTC.*

139-149 If you cannot, or decide not to, run a regional model, I think you should provide more justification for choosing to run this particular version of ECHAM, especially since you kind of denigrate its performance later in the manuscript.

To the best of our knowledge, the ECHAM-HAM is the only model with a charcoal module implemented.

*L142-143: To the best of our knowledge, this is the only model with a charcoal module implemented.*

151-157 Curious why you chose to only use MODIS products. It is becoming increasingly clear that important details are missed due to coarse spatial resolution, and the fixed single overpass time. Are there not relevant products from the Sentinel satellites? Geostationary platforms (mainly supporting meteorology forecasting) can provide insight throughout the day, especially later in the afternoon when many fires are strongest.

We analysed higher spatial resolution Soumi/VIIRS active fire products (375m) next to MODIS active fire products to obtain a better understanding on the spread and evolution of the fire (see also answer to the comment of Paulo Fernandes). To be consistent with the emission estimates of the FINN v1.6 database (based on MODIS active fire data), we decided here to show MODIS fire products. To make this clearer we rephrased it in the manuscript.

*L157-159: In consistency with FINN v1.6, we chose the Thermal Anomalies & Fire Daily L3 Global Product, which is also utilized for the emission model. In addition, we obtained the Burned Area Monthly L3 Global Product of MODIS.*

184 I would not say that peak rBC of 9.8 ng/g is "remarkable". It is only about 2 x higher than the secondary peak at 60-70 cm depth. The Thomas et al., 2017 paper cited elsewhere found the average peak in 22 north Greenland pits to be 15 ng/g, with max of 43 ng/g with longer transport distances back to the source fires.

Agreed, remarkable was deleted.

193-194 I would mention the secondary rBC peak in samples 12/13 that overlaps the bump up in charcoal sample 5, especially since you point out the increased size later to suggest more local source.

Good point. We now mention this secondary rBC peak.

*L197-200: Below the rBC peak, a secondary rBC maximum with 4.5 ng g$^{-1}$ was observed between 60 and 70 cm in one replicate series, but otherwise average concentrations are low (2.0 ng g$^{-1}$). Except for this secondary maximum, a very good agreement is obtained between the two series of replicate rBC samples (r = 0.90, all samples).*

197 Regarding the "narrower" charcoal peak, I think it may mostly be in the 2 cm interval 28-30 cm below rBC sample 6, but that is just a hypothesis.

That seems likely, but we cannot proof it with our data. We added this sentence.

*L239-240: Dry deposition most likely resulted in a confined charcoal layer, which was not separated from the rBC peak in the snowpit due to the coarse sampling resolution of 10 cm.*

270-280 As noted above, consider forward trajectories from the fires, specifically looking to see if smoke was likely over JFJ when it started snowing 25 June. And seriously consider whether ECHAM is possibly correct that the BC in the snow is not from the fires.

See comment above.

Section 3.5 See earlier question about exclusive reliance on MODIS, and the on-line comment from Paolo Fernandes.

See above reason for using MODIS and our response to the comment from Paolo Fernandes.

Section 3.6 Consider the "deposited" BC and charcoal calculations presented in introductory comments. If you decide to stick with flux estimates provide more details about assumptions used to get values so much higher than the model.

Calculations were changed, see comment above.

Last paragraph of this section, seems that the model thought there was no smoke at all over JFJ, pointing to a transport shift (or error) rather than problems with emissions. I made a case that the model came within factor of 2.5 of the total amount of BC in the snow from 25-26 June, even though it wants to deposit most of it on the 27th and 28th . Might be worth comparing the timing and amount of precipitation in the model to observations.

We extracted only deposition from the model. Since the model obtained the charcoal deposition peak, it must have seen the smoke over JFJ. See also our above comment on the fluxes.

**Anonymous Referee #2**

This manuscript attempts to link emissions from a known fire event and their deposition on snow close to JFJ in the Swiss Alps through extensively investigating a severe fire event occurred on 17-24 June 2017 in Portugal. This study is attempting to provide an interesting approach, connecting a set of valuable record of charcoal, black carbon, and ions in the snow pit and a combination of atmospheric in-situ measurement, remote sensing, air mass trajectory calculation, and transport simulation, to better understand the information of particle deposition in snow in European high-altitude sites. This approach is very useful to understand atmospheric processes of aerosol particles how to transport over long distances, be scavenged by snow fall, and be deposited to snow by providing clues. However, there is an issue that should be considered seriously to publish the results. Authors measured charcoal, refractory black carbon (rBC), and ions in snow collected near JFJ and compared their profiles with equivalent black

carbon (eBC) measured in the atmosphere at JFJ. The elevations of charcoal in the upper layer of C in snow (sample "2" in figure 2e) and eBC in atmosphere on 22 June (figure 2a) were obvious with an increase relative to background level by _6 times and _10 times, respectively, indicating that the fire plume reached JFJ. In contrast to them, rBC in sample "4-6" (figure 2d) increased a little and changes in ion concentrations are even indiscernible. Readers reasonably expect that potassium, ammonium, and nitrate can be elevated in their concentrations if the big fire plume indeed reached and is detectable by charcoal and eBC. In the same snow layer, the rBC concentration was elevated just 1.5 times relative to the second peak with rBC concentration of _5 ng/g, that is probably from local fires as suggested by authors. Discuss this issue in dept and provide more evidences and/or assumption to support your argument.

This is a really helpful suggestion, and the point was raised also by referee 1. We were from the beginning puzzled by not finding elevated concentrations of the other fire tracers (ammonium, formate, and acetate) in the samples 4-6. Re-considering this in view of both referee's comments, we revised our interpretation. Charcoal originating from the Portugal fires was deposited by dry deposition during the period 22$^{nd}$-24$^{th}$ June, when the fire plume was detected by atmospheric eBC measurements at JFJ. Since no snowfall occurred during that period, the majority of BC was not deposited, since dry deposition is not efficient for submicron particles. Dry deposition of charcoal most likely resulted in a confined layer, which was not resolved with the 10 cm sampling resolution. With the snowfall on 25$^{th}$ June, air mass transport changed, ending the advection of the fire plume to the JFJ as indicated by the backward trajectories arriving at JFJ at 18 UTC (will be included in Fig. 4 in the revised version). Instead more regional polluted air masses were scavenged. That the charcoal and rBC peaks partly overlap is most likely due to the coarse resolution of the charcoal samples. The change of air masses does explain the lack of a peak in the other fire tracers, e.g. ammonium, and the time delay between the charcoal and the BC peak in the model output and the snow pit data. The manuscript was changed accordingly.

*Abstract:*

*L26-38: According to modelled emissions of the FINN v1.6 database, the fire emitted a total amount of 203.5 tons BC from a total burned area of 501 km$^2$ as observed on the basis of satellite fire products. Backward trajectories unambiguously linked a peak of atmospheric equivalent BC observed at the Jungfraujoch research station on 22$^{nd}$ June, with elevated levels until the 25$^{th}$ June, with the highly intensive fires in Portugal. The atmospheric signal is in correspondence with an outstanding peak in microscopic charcoal observed in the snow layer, depositing nearly as many charcoal particles as during an average year in other ice archives. In contrast to charcoal, the amount of atmospheric BC deposited during the fire episode was minor due to a lack of precipitation. Simulations with a global aerosol climate model supported that the observed microscopic charcoal particles originated from the fires in Portugal and that their contribution to the BC signal in snow was negligible. Our study revealed that microscopic charcoal can be transported over long distances (1500 km), and that snow and ice archives are much more sensitive to distant events than sedimentary archives, for which the signal is dominated by local fires. The findings are important for future ice-core studies, as they document that for BC as fire tracer the signal preservation depends on precipitation. Single events, like this example, might not be preserved due to unfavorable meteorological conditions.*

*Fire tracers:*

*L237-241: We hypothesize that charcoal originating from the Portugal fires was deposited by dry deposition during the period $22^{nd}$ to $24^{th}$ June, when the fire plume arrived at the JFJ as detected by elevated atmospheric eBC concentrations. Since no snowfall occurred during that period, the majority of BC was not deposited. Dry deposition most likely resulted in a confined charcoal layer, which was not separated from the rBC peak in the snowpit due to the coarse sampling resolution of 10 cm. With the beginning of snowfall on $25^{th}$ June, air mass transport changed, ending the advection of the fire plume to the JFJ as indicated by backward trajectories (see below and Fig. 4). Instead more regional polluted air masses were scavenged, which explains the absence of ionic fire tracers and of a shift in the rBC size distribution.*

*Atmospheric transport:*

*L271-274: The back trajectories with arrival at JFJ on $25^{th}$ June at 18 UTC indicates a major change in the synoptic situation to more north-westerly flow directions. We can only speculate that this happened concomitant with the onset of precipitation, since the timing of the latter is not precisely known.*

*L279-283: Furthermore, other BC sources than fires seem to dominate in our simulations, which is in agreement with our hypothesis that during fire plume arrival at JFJ the majority of fire-related BC was not deposited due to the lack of snowfall. With the beginning of snowfall at JFJ, air mass transport changed and more regional polluted air masses with minor or without fire contribution were scavenged.*

*Conclusions:*

*L349-350: Dry deposition of microscopic charcoal resulted in an outstanding peak in the snowpack. This event deposited nearly as many charcoal particles as during an average year in other ice archives.*

*L354-363: For rBC, in contrast, concentrations in the snow were not exceptionally high. In combination with the absence of a peak in ionic fire tracers such as ammonium, this suggest that the majority of atmospheric BC was not deposited during the fire episode due to a lack of precipitation. Instead the observed rBC peak was mostly likely caused by scavenging of air masses containing regional pollution with the beginning of snowfall on $25^{th}$ June, which ended the advection of the fire plume to the JFJ. rBC scavenging ratios were in line with previous studies, giving additional evidence that rBC was predominantly scavenged by wet deposition. Simulations with a global aerosol climate model supported that the observed microscopic charcoal particles originated from the fires in Portugal, whereas their contribution to the BC signal in snow was minor. The findings of our case study are important for future ice-core studies, as they document that for BC as fire tracer the signal preservation depends on precipitation and wet deposition. Single events, like this example, might not be preserved due to the unfavorable meteorological conditions.*

L135: Since the fire event studied in this work was a well-documented recent extreme event, authors need to calculate forward air mass trajectories to see if the plume departing from the exact fire spot reach the JFJ site. Consider that a starting height of air mass trajectories is not necessarily just above ground because fire-emitted particles can be directly injected into the free troposphere up to > 3 km. Also, it would be

great to see if the area-averaged time series of AOD match with forward air mass trajectories. For AOD, just simply check GIOVANNI platform (https://giovanni.sci.gsfc.nasa.gov/giovanni/#service=DiArAvTs&starttime=&endtime=&bbox=- 180,- 90,180,90). Intense fire plumes are often easily captured by AOD, and thus AOD near JFJ should be elevated day-by day.

We calculated forward trajectories as suggested by the reviewer and they basically agree with the backward trajectories. We prefer to keep the backward trajectories in the manuscript, since they are more conclusive, but added the one arriving at JFJ on 25$^{th}$ June 18 UTC to show the change to north-westerly air flow. We show now trajectories calculated 5-days backward at 20 equidistant levels in pressure coordinates between 700 hPa and 500 hPa, in accordance with the detected smoke plume height.

Thanks for the suggestions of the GIOVANNI database to explore changes in AOD concentrations. We had a look into different daily AOD products (e.g. from OMAEROe v003 at 0.25° (smallest spatial resolution in the GIOVANNI database) to the MODIS-Terra MOD08_D3 v6.1. at 1° spatial resolution) and their area-averaged time series for 17$^{th}$ until 26$^{th}$ of June. However, this didn't help to address your question because 1) of their rather coarse spatial resolution, 2) cloud coverage which obscured the satellite's view. The data are inclusive because of their poor spatial resolution.

In addition, we looked at AOD data with a spatial resolution of 10 km at nadir from the Modis sensor flying onboard of the satellites Terra and Aqua, which support that between 20$^{th}$ and 22$^{nd}$ of June the JFJ site received air masses with elevated AOD concentrations from Portugal. The AOD concentration levels increased over Switzerland mainly North of the Alps during that time, in agreement with the backward trajectories (Fig. 4). From 23$^{rd}$ to 25$^{th}$ of June the area was cloud covered, so AOD could not be retrieved.

This was included in the manuscript.

*L172-175: To detect the fire plume, we analyzed different daily Aerosol Optical Depth (AOD) products from OMAEROe v003 at 0.25° (smallest spatial resolution in the GIOVANNI database), MODIS-Terra MOD08_D3 v6.1. at 1° spatial resolution, and from the Modis sensor flying onboard of the satellites Terra and Aqua at a spatial resolution of 10 km at nadir and their area-averaged time series for 17$^{th}$ until 26$^{th}$ of June.*

*L274-277: AOD data at 10 km resolution support that between 20$^{th}$ and 22$^{nd}$ of June the JFJ site received air masses with elevated AOD from Portugal. The AOD levels increased over Switzerland mainly north of the Alps during that time, in agreement with the backward trajectories (Fig. 4). From 23$^{rd}$ to 25$^{th}$ of June the area was cloud covered, so AOD could not be retrieved.*

L206: The upper size limit of BBHG is generally <300 nm, although which depends on the instrumental setting. Please check if authors mentioned "BBLG" instead of "BBHG".

Thank you for discovering this! The sentence was changed accordingly.

*L209-210: The mass size distribution for the broadband low gain (BBLG) channel of the SP2 remained similar throughout the profile with 306 nm and 291 nm for the two series of replicates, respectively (Fig. 2d).*

L205-214: In figure 2d, the diameter of rBC particles in sample "4-6" (figure 2d) seem similar to that of other layers except for sample "13". The size of atmospheric rBC particles generated from biomass combustion is generally larger than that from urban fossil emissions. If the rBC size is not obviously large in sample "4-6", the possible reason should be suggested, regarding for example, cloud or wet-scavenging during transport and/or scavenging by snowfall, etc.

This finding fits now better with our revised interpretation, see discussion above, and changed the discussion about the size distribution.

*L211-212: Only sample 13 shows a bigger fraction of large rBC particles, in association with a small peak in rBC and charcoal concentration. On the contrary, no shift in the rBC size distribution is visible for samples 4 to 6 during the peak in rBC and charcoal concentrations.*

L224: Authors mentioned that smoke particles can be lifted up to free troposphere and travel over long distance. It can be true not only for charcoal but also for rBC particles. As noted above, it should be seriously considered why rBC and ionic particles did not elevated unlike charcoal. rBC particles are small with diameters < 1 micrometer, which thus can travel longer distances, as found in previous studies so far. Also, authors may need to check if the number of particles in SP2 scattering channel ("SCLG" or "SCHG") is elevated in the snow layers. The profile of scattering particles, i.e., number concentration, might correspond to that of charcoal.

We agree. rBC particles should have been transported as well. We think that charcoal was deposited by dry deposition whereas rBC was not scavenged until $25^{th}$ June due to the lack of precipitation, see comment above. We revised our interpretation accordingly, see comment above.

Charcoal particles detected microscopically had a major axis >10 μm. Such large particles are not aerosolized with the nebulizer set-up and, thus, do not enter the SP2. In addition, it is not possible to analyse insoluble scattering-only particles in aerosolized snow (or other liquid) extracts since extremely large numbers of scattering particles are produced by aerosolization and drying of the dissolved components in the solution. The numbers of such particles overwhelm any SP2 scattering signals from the insoluble components.

L230: Both ammonia (NH3) and NOx can be emitted from fires, and during the transport time in the atmosphere, ammonium nitrate (NH4NO3) and other forms of nitrate and ammonium can be formed particularly under high relative humidity via aqueous reaction. Authors should seriously consider why such inorganic ions are not sufficiently detected in the snow layer corresponding to smoke plume. Meteorological conditions, for example too dry condition in FT and/or too warm in PBL, were not favorable for the aerosol formation? Moreover, potassium has been broadly used as an indicator of biomass combustion so far. Did authors observe a peak of potassium in the snow layer?

We agree and revised our interpretation accordingly, see above.

**Paulo Fernandes**

pamfernand@gmail.com

450

This is interesting work. Just a short comment in relation to the burned area estimate. The fire blowup, with extremely high rate of spread and pyCb formation and collapse that killed 66 people, occurred on the first day of its development. As such your 17th June burned area estimates are underestimated by one order of magnitude. According to our reconstruction (CTI 2017), resulting from the combination of

455    various ground- and remote sensing-based information, burned area on the 1st and 2nd days was 128 and 211 km2, respectively. Thus remote sensing products did not detect fire growth nor peak FRP happening at about 19-21 h PM on the 17th, presumably because of the combination of dense smoke with thunderstorm clouds.

Thank you for your comment. We actually analysed remote sensing data for the 17th, but did not

460    describe this in detail in the manuscript. On the 17th, the fire was early detected by Soumi/VIIRS (375m) about 13:46-13:47 UTC few kilometres from a thick cumulonimbus cloud. For Aqua/MODIS 1km data (overpass time 13:45-13:50 UTC – almost simultaneous overflight), the fire was too small to be detected. This explains why FINN has no emission entries for the 17th. This was included in the manuscript.

465    *L298-302: An additional comparison of the Aqua/MODIS and VIIRS/NPP Active Fire Product (375 m spatial resolution; Schroeder at el., 2014) showed, that the fire was early detected by the VIIRS/NPP Fire Product at ~13:46 UTC. The fire was located few kilometres away from a thick cumulonimbus cloud. Though, the overpass time of Aqua/MODIS was almost simultaneous (i.e. 13:45-13:50 UTC), the fire was too small to be detected by the coarser spatial resolution sensor (1 km). This explains why*

470    *FINN v1.6 has no emission entries for 17th June.*

**Response to comments from the editor**

Nevertheless, I will be greatly appreciate in the revised version if the authors can give the definition of eBC and rBC at first occurrence (abstract and text), referring to a published article is not enough.

475    Good comment, thank you. We included the definition for both in the text, but not in the abtract.

*L90-92: The term eBC is used for black carbon data derived from optical absorption methods, together with a suitable mass absorption cross-section (MAC) for the conversion of light absorption coefficient into mass concentration (Petzold et al., 2013).*

*L123: rBC stands for black carbon measured by incandescence methods (Petzold et al., 2013)*

480    Also, I will add a comment in the discussion as I'm very critical about the use of the scavenging ratio.

We agree with that comment, see also our comment to referee 1. Nevertheless, despite all uncertainties included, scavenging ratios give at least some indication which components are easily taken up by snow. Since there are so few data we preferred to keep them.

Also regarding the comment on AOD by review 2, even if the comparison is not fully conclusive, I suggest
to mention it in your MS. I will avoid other readers to ask themselves the same question.

This was included in the manuscript.

*L172-175: To detect the fire plume, we analyzed different daily Aerosol Optical Depth (AOD) products from OMAEROe v003 at 0.25° (smallest spatial resolution in the GIOVANNI database), MODIS-Terra MOD08_D3 v6.1. at 1° spatial resolution, and from the Modis sensor flying onboard of the satellites Terra and Aqua at a spatial resolution of 10 km at nadir and their area-averaged time series for 17th until 26th of June.*

*L274-277: AOD data at 10 km resolution support that between 20th and 22nd of June the JFJ site received air masses with elevated AOD from Portugal. The AOD levels increased over Switzerland mainly north of the Alps during that time, in agreement with the backward trajectories (Fig. 4). From 23rd to 25th of June the area was cloud covered, so AOD could not be retrieved.*

Same thing for the comment of reviewer 1 for the use of the CHAM-HAM model, Please, justify in the text why you decided to use this global model instead of a regional model.

We included that in the manuscript.

[revised manuscript text omitted]